# The relation between Rossby Wave Breaking events and low-level weather systems

Talia Tamarin-Brodsky[1] and Nili Harnik[1]

[1]Department of Geophysics, Tel-Aviv University, Tel-Aviv, Israel

**Correspondence:** Talia Tamarin-Brodsky (taliatamarin@gmail.com)

**Abstract.** Rossby wave breaking events describe the last stage in the life-cycle of baroclinic atmospheric disturbances. These breaking events can strongly influence the large-scale circulation and are also related to weather extremes such as heat waves, blockings, and extreme precipitation events. Nonetheless, a complete understanding of the synoptic-scale dynamics involved with the breaking events is still absent. For example, it is not clear how well the theoretical life-cycle experiments, which use a specified initial perturbation with a single zonal wavenumber and a prescribed simplified initial zonal jet, capture the life-cycle of real-atmosphere weather systems. Here we combine a storm-tracking technique together with a wave breaking detection algorithm, to examine how upper level wave breaking and surface weather systems are related in the North Atlantic during winter. These datasets allow us to examine whether upper-level wave breaking and low-level weather systems always occur simultaneously, and if we can identify preferred relations between the surface weather system type (cyclone or anticyclone) and the type of the upper-level breaking event (cyclonic or anticyclonic wave breaking denoted CWB or AWB, respectively). We find that in the North Atlantic, most weather systems are associated with an AWB and/or CWB at some point during their lifetime, while only few cyclones and anticyclones do not involve any upper-level wave breaking (roughly 11% and 15%, respectively). Our results imply that composites of cyclones and anticyclones involve a mixture of different types of life-cycles, depending on whether they involve CWB or AWB, as well as their position relative to the RWB center. Moreover, the system characteristics (including actual and relative positions, intensities, and displacements) differ depending on the associated breaking type. We distinguish between "same-pairing" cases (i.e., cyclones with CWB and anticyclones with AWB) and "opposite-pairing" cases (i.e., cyclones with AWB and anticyclones with CWB). Compositing the cyclones and anticyclones based on this criterion, we find that in similar-pairings the surface system is positioned so that its associated upper-level winds would enhance the breaking (the anomalous circulation is in the same direction as the background shear), but for opposite-pairings, the upper-level winds associated with the surface system do not act to enhance the breaking which occurs in the direction of the background shear. A better understanding of the different life-cycles of real-atmosphere cyclones and anticyclones and the upper-level breaking they involve is important for exploring the relation between storm-tracks and slowly varying weather regimes and how it is mediated by RWB events.

# 1 Introduction

The midlatitude atmospheric circulation is characterized by the continuous passage of propagating synoptic-scale weather systems, which play an important role in the meridional redistribution of momentum, moisture, and heat. These anomalous cyclonic and anticyclonic circulations grow via baroclinic instability, as a result of Earth's rotation and the equator-to-pole temperature difference. The waves tend to follow a typical Rossby wave life-cycle, which involves a linear baroclinic growth stage (Charney, 1947; Eady, 1949) followed by a nonlinear barotropic decay (Simmons and Hoskins, 1978; Davies et al., 1991).

During the decay stage, Rossby wave breaking (RWB) occurs, formally defined as a large-scale and irreversible overturning of the potential vorticity (PV) contours on isentropic surfaces (McIntyre and Palmer, 1983). This overturning results in an inversion of the meridional PV gradient, which is often used as a definition of RWB in automated detection algorithms. The PV mixing that occurs in the wave breaking region, and the associated anomalous momentum fluxes, can result in an acceleration/deceleration or meridional shifts of the upper-level jet. In addition, these breaking events were linked to extreme

weather events (Martius and Rivière, 2016), such as extreme precipitation (Moore et al., 2019; de Vries, 2021), explosive cyclogenesis (Hanley and Caballero, 2012; Gomara et al., 2014), and tropical cyclone activity (Zhang et al., 2017; Zhang and Wang, 2018).

There are two main types of wave breaking, with a very distinct upper-air behaviour, occurring at the end of the baroclinic wave life-cycles (Simmons and Hoskins, 1978; Thorncroft et al., 1993). The first type is dominated by an Anticyclonic Wave

Breaking (AWB) (e.g., Fig. 1a), and is characterized by a southwest-northeast (SW-NE) tilt of the PV contours. The second type involves Cyclonic Wave Breaking (CWB) (e.g., Fig. 1b) and is characterized by a southeast-northwest (SE-NW) tilt of the PV contours. Previous studies have further distinguished between 'equatorward breaking' and 'poleward breaking' cases (e.g., see Fig.1 in Tyrlis and Hoskins (2008) or Fig.1 in Gabriel and Peters (2008) for a schematic illustration). The equatorward breaking cases involve an equatorward extrusion of high-PV air, and were coined LC1 and LC2 by Thorncroft et al. (1993).

In LC2, CWB occurs on the poleward side of the jet where the shear is cyclonic, which leads to the development of wide and strong troughs and prevents the building of strong ridges. In LC1, AWB occurs on the equatorward side of the jet where the shear is anticyclonic, leading to thinning troughs and possibly to the development of a weak cutoff low (Thorncroft et al., 1993). The importance of the two poleward breaking cases (P1 and P2), which involve a poleward extrusion of low-PV air, was introduced by Peters and Waugh (1996). In P2, AWB occurs on the equatorward anticyclonic side of the jet, which results

in strong blocking-like ridges that tend to develop equatorward of the jet. In P1, CWB occurs on the poleward cyclonic side of the jet, leading to thinning ridges and often to the development of a weak cutoff ridge (Peters and Waugh, 1996). Hence, P2 and LC1 are dominated by AWB, while P1 and LC2 are dominated by CWB. In all four cases, a reversal in the upper-level meridional PV gradient is observed, but the overturning of the PV contours is weaker in cases P1 and LC1. Most traditional RWB-detection algorithms that rely on the reversal of PV contours only distinguish between AWB and CWB events, and in

practice mostly detect the P2 and LC2 types. This is often not acknowledged in studies, where the more familiar LC1 and LC2 life-cycle terminology is used instead. A further investigation of each one of the four breaking types was performed in several studies (e.g., Peters and Waugh, 1996; Tyrlis and Hoskins, 2008; Gabriel and Peters, 2008).

Past studies have shown, in idealized baroclinic life-cycle experiments, that the sense in which the wave breaking occurs (AWB or CWB) can be controlled by changing either the initial meridional shear of the background zonal jet (Simmons and Hoskins, 1978; Davies et al., 1991; Thorncroft et al., 1993; Peters and Waugh, 1996; Hartmann and Zuercher, 1998; Hartmann, 2000; Shapiro et al., 1999), the initial zonal wavenumber of the perturbation (Hartmann and Zuercher, 1998; Orlanski, 2003; Wittman et al., 2007), the strength of the noise added to the initial perturbation (Jäger et al., 2023), or the strength of the cyclonic and anticyclonic vortices, achieved by adding external forcing or by adding moisture (Orlanski, 2003). The type of breaking can significantly modify the low-frequency atmospheric circulation, and therefore influence the jet variability. In general, during AWBs eddy momentum fluxes $u'v'$ are mostly poleward, and the zonal flow is therefore accelerated (decelerated) poleward (equatorward) of the breaking, leading to a poleward shift of the jet. On the contrary, during CWBs momentum fluxes are mainly equatorward, and thus accelerate (decelerate) the zonal flow equatorward (poleward) of the breaking, leading to an equatorward shift of the jet (Simmons and Hoskins, 1978; Thorncroft et al., 1993). Similar conclusions were reached from a vorticity flux perspective by Orlanski (2003), who suggested that when anticyclonic circulations are dominant, the eddy vorticity flux $v'q'$ (where $q'$ is the relative vorticity anomaly) is positive poleward of the breaking and negative equatorward of it, which acts (through $\frac{\partial \overline{U}}{\partial t} \sim \overline{v'q'}$) to accelerate the zonal flow poleward of the breaking and decelerate it equatorward of it (and vice versa for the case where cyclonic circulations are dominant, see also Fig. 9 in Orlanski, 2003)

The interaction between the shorter-timescale RWB and the lower-frequency background flow is two-way. On the one hand, the low-frequency patterns of variability were shown to influence the type and frequency of RWB. For example, the frequency of AWB and CWB can be modulated by midlatitude weather regimes (Franzke et al., 2011; Swenson and Straus, 2017), the El Niño Southern Oscillation (ENSO) (Waugh and Polvani, 2000; Shapiro et al., 2001) or the Madden-Julian Oscillation (MJO) (Cassou, 2008). On the other hand, RWBs were shown to modify the low-frequency variability by triggering or extending the life-time of weather regimes (Michel and Rivière, 2011; Michel et al., 2012), and has an important role in the onset and decaying stages of blockings (Hoskins et al., 1983; Tyrlis and Hoskins, 2008; Woollings et al., 2008; Tyrlis and Hoskins, 2008; Woollings et al., 2011; Masato et al., 2012). More generally, a positive feedback was identified between RWBs and the latitudinal position of the jet, as a more poleward (equatorward) jet implies that AWB is more (less) probable (and vice versa for CWB), hence the jet is pushed or maintained further poleward (equatorward) by the eddy forcing (Rivière, 2009).

Moreover, it was suggested that the North Atlantic Oscillation (NAO), the leading mode of winter low-frequency variability in the North Atlantic region, can be viewed as variations in the frequency, type, and location of RWB events (Benedict et al., 2004; Franzke et al., 2004; Rivière and Orlanski, 2007; Woollings et al., 2008; Strong and Magnusdottir, 2008; Kunz et al., 2009). Generally, these studies find that AWBs (CWBs) are associated with the positive (negative) polarity of the NAO. Woollings et al. (2008) suggested that the negative NAO can be viewed as a period of more frequent high-latitude blocking events associated with CWB, resulting in a more zonal and southward jet regime, while the positive NAO can be viewed as period in which these events are infrequent, resulting in a more tilted and northward jet regime.

In addition, several previous studies have examined composites of RWB events. For example, Strong and Magnusdottir (2008) looked at composites of AWB and CWB events in the Northern Hemisphere (NH), and found that AWBs are associated with a negative Sea Level Pressure (SLP) anomaly poleward of the breaking and a positive SLP anomaly equatorward of the

breaking center, similar to the SLP signature of the positive NAO (and the opposite for CWB). Similar results were found by Kunz et al. (2009), who examined upper and lower tropospheric composites of RWB events in a simplified General Circulation Model (GCM), to study the potential of AWB and CWB events to drive NAO-like meridional circulation dipoles. Finally, Zhang and Wang (2018) examined composites of AWB in the North Atlantic during the warm season, to highlight the role of diabatic heating in contributing to the wave breaking.

Another motivation to study upper level RWB events has been their strong connection to the evolution of low-level cyclones and anticyclones. For example, various studies examined the synoptic-scale evolution and baroclinic life-cycle of cyclones, which inherently involve the evolution of the upper-level trough (e.g., Shapiro and Granas, 1999). Previous studies have shown how a precursor wave breaking can influence the cyclone's formation. For example, PV streamers associated with AWB are often found as precursors to subtropical, tropical, and Mediterranean cyclones (e.g., Appenzeller et al., 1996; Davis, 2010; Galarneau et al., 2015; Flaounas et al., 2015; Bentley et al., 2017; Portmann et al., 2021; Flaounas et al., 2022). For midlatitude cyclones, the existence of simultaneous AWB and CWB events in the eastern North Atlantic was shown to lead to a stronger and more zonally extended upper-level jet, which favors the formation of explosive storms reaching central Europe (Hanley and Caballero, 2012; Messori and Caballero, 2015). Michel et al. (2012) analyzed the link between surface cyclones and upper-tropospheric Rossby wave breaking during the Scandinavian Blocking (SB) regime. They found differing cyclone trajectories, associated with different types (cyclonic/anticyclonic) of wave breaking occurring during the onset and decay of the SB regime. In addition, Gomara et al. (2014) examined the two-way relationship between RWB and explosive cyclones over the North Atlantic, and found that the latter are associated with enhanced frequency of RWBs several days prior to the cyclone's maximum intensification. They also found some signature for enhanced occurrence of CWB over southern Greenland and AWB over Europe after explosive cyclogenesis, but only for very intense cyclones.

Taken together, the above studies suggest an underlying picture in which low-frequency large-scale atmospheric weather regimes, typically defined via the mid or upper level flow, interact with surface weather systems via RWB events. A better understanding of this interaction, as it manifests in the real atmosphere, is needed in order to improve our understanding of the processes shaping the large-scale distribution of weather extremes (e.g., Coumou et al., 2014; Hoskins and Woollings, 2015; Kautz et al., 2022), subseasonal weather predictability (e.g., Mariotti et al., 2020), and projected future circulation changes (e.g., Woollings et al., 2008; Coumou et al., 2018). The observed relationship between RWB and weather regimes is relatively well-established based on statistical analyses of multiple events, especially in the North Atlantic (Strong and Magnusdottir, 2008; Michel and Rivière, 2011; Swenson and Straus, 2017). However, our understanding of the relationship between RWB and weather systems is mostly based on idealized studies (e.g., Simmons and Hoskins, 1978; Davies et al., 1991; Thorncroft et al., 1993) or single-event case studies (e.g., Shapiro and Granas, 1999). Thus, a similar comprehensive picture of the relation between RWB and weather systems does not exist in the literature, to the best of our knowledge. This is the aim of the current study. We address this gap by combining a Lagrangian feature-tracking technique (to identify and track low-level cyclones and anticyclones) together with a wave-breaking detection algorithm (to identify the times and positions of cyclonic and anticyclonic RWB events). Specifically, we aim to address the following open questions:

1. What is the relation between RWB events and low-level weather systems? For example, do RWBs and weather systems always occur simultaneously? What are the percentages of cyclones and anticyclones involved with each type of breaking, and vice versa?

2. How do the weather system characteristics (including geographical positions, intensity, and displacements) and composite time evolution differ, depending on the type of upper-level RWB and their position relative to the breaking?

3. To what extent are the life-cycles of real-atmosphere cyclones and anticyclones captured by the existing idealized life-cycle experiments (e.g., Simmons and Hoskins, 1978; Davies et al., 1991; Thorncroft et al., 1993), which use a specified initial perturbation with a single zonal wavenumber and a prescribed simplified initial zonal jet?

While examining case studies can be very insightful, some of these questions cannot be addressed based on individual cases alone. Using automated detection algorithms of RWBs and weather systems can give a more comprehensive picture of possible cases, and therefore supplement existing studies and generalize their results.

The paper is organized as follows. Section 2 reviews the data and methods used for the analysis, including the RWB detection algorithm, the Lagrangian storm-tracking technique, and the compositing procedure. In section 3, wave breaking-centered analysis is performed, and the fundamental relation between upper-level RWB events and low-level weather systems in the North-Atlantic is presented, which shows the different characteristics of cyclones and anticyclones involved with AWB and CWB events. Section 4 examines the life-cycle of the weather systems in different RWB types, which is divided into "same-sense" weather system vorticity and RWB type (i.e., anticyclones during AWB and and cyclones during CWB), and "opposite-sense" weather system vorticity and RWB type (i.e., anticyclones during CWB and and cyclones during AWB). Conclusions are discussed in section 6.

## 2   Data and methods

In this study we use the six-hourly upper-level (250 hPa) horizontal velocities and Potential Vorticity (PV), PV on the 350K isentropic level, SLP, and lower-level (850 hPa) horizontal wind and vorticity, from the European Centre for Medium-Range Weather Forecasts (ECMWF) interim reanalysis dataset (ERAI; Dee et al., 2011). The data covers the years 1980-2014 during the NH winter (December-January-February, DJF) period. Climatology is defined as the winter-average over these 35 years, while anomalies are defined as deviations from the DJF climatology. We focus our analysis on the North Atlantic region, defined here as (30N-60N, 80W-20E). Note that the analysis is performed on the ERAI reanalysis data rather than on the newer ERA5 dataset for consistency between existing analyses of the tracking and wave-breaking detection results. However, we do not expect any major or fundamental differences in our conclusions if the ERA5 dataset was used instead.

### 2.1   Lagrangian storm-tracking algorithm

For the tracking of the low-level cyclones and anticyclones we use the objective feature-tracking algorithm TRACK of Hodges (1995, 1999), which is a widely used storm-tracking algorithm. We use the 850 hPa relative vorticity field, and the cyclone

and anticyclone centers are then identified by a local maximum and minimum of the vorticity field, respectively. The intensity is determined based on the relative vorticity anomaly (in absolute value) at the center of the system, with a cutoff value of $10^{-5}$ s$^{-1}$ for the identification of the weather system (a threshold customarily used for the identification of cyclones and anticyclones). The background flow is automatically removed by the algorithm prior to the tracking by subtracting all spatial wavenumbers smaller than or equal to 5, to isolate the synoptic-scale features. In addition, the vorticity field is reduced to a T42 grid, and then a spectral tapering is performed in order to suppress Gibbs phenomenon (Hodges, 1995). The centers of weather systems are tracked every six hours, and the tracking is performed on the sphere, by first initializing the maxima or minima into a set of tracks using a nearest neighbour method, and then refining these by performing a constrained minimization of a cost function for track smoothness (Hodges, 1999). The tracking is performed separately for cyclones and anticyclones, for each winter during the years 1980-2014 (where the year is defined according to January).

## 2.2 Rossby wave-breaking detection algorithm

Here we use the RWB detection algorithm developed by Strong and Magnusdottir (2008), which can detect both poleward and equatorward breaking events. We configure it to detect the equatorward-breaking high-PV tongues associated with anticyclonically and cyclonically overturning PV contours, but our results are qualitatively similar if poleward breaking cases are analyzed instead. The RWB distributions depend on the vertical isentropic level chosen, with generally more frequent AWB and less frequent CWB at higher isentropic levels (e.g., Martius et al. , 2007). Here we use PV on the 350K isentropic level (corresponding approximately to the upper troposphere/lower stratosphere) for the identification of RWB events, which has a relatively strong RWB activity. As noted in earlier studies who used the same RWB detection algorithm (e.g., Strong and Magnusdottir, 2008; Zhang and Wang, 2018), the 350K level provides a useful representation of both AWB and CWB events over all latitudes, because higher latitude RWB events are deep enough to be detected by higher PV values at this level. Hence, the RWB identification is performed for each PV value between 1.5-7.5 PVU, which allows for detecting RWB in both lower and higher latitudes. Note that similar results were also found by using PV on the 250 hPa level (not shown).

The algorithm identifies large-scale overturning of circumpolar PV contours, on each one of the PV contours, by searching for contours crossing a particular meridian more than once. The algorithm uses the geometry of the overturning PV contour to quantify the zonal extent of the break and to distinguish between anticyclonic and cyclonic overturning (see Strong and Magnusdottir, 2008 for more details). In the original algorithm, the center of the breaking event is defined as the geographic centroid of the PV tongue (the equatorward PV tongue in our case). This is slightly modified here such that the latitudinal center of the event is chosen between the poleward and equatorward PV tongues (for presentation purposes mainly). The latter is achieved by adding half of the meridional width of the tongue (at its centroid longitude) to the centroid position. For each breaking event, only the spatially largest overturning is taken among the 1.5-7.5 PVU contours, and further events occurring less than $L_x$ apart ($L_x$ being the tongue's extent) are eliminated. In addition, if RWBs occur on adjacent days, only the day of maximum overturning (spatially largest overturning) is considered. The longitudinal width of an AWB (CWB) overturning is set to be larger than 7° (5°), the area of the breaking (calculated as a fraction of earth's surface area) larger than $7 \cdot 10^{-4}$, and the depth of the breaking (defined as the maximum PV value in the tongue minus the analyzed PV contour) should be larger

than 1 PVU. A different threshold is used for longitudinal width of the AWB and CWB events, since the former are generally larger and greater in number, but similar qualitative results are obtained for other thresholds. Finally, only RWB events whose centroid lies within the Euro-Atlantic domain, defined here as the box (15N-75N, 80W-20E) are considered.

Examples of AWB and CWB events detected by the algorithm are shown in Fig. 1a and Fig. 1b, respectively. As can be seen in these examples, AWB (Fig. 1a) involves a high-PV streamer wrapping anticyclonically around a low-PV ridge. The AWB centroid is located in the equatorward (i.e. anticyclonic) side of the upper-level zonal flow, and the orientation of the PV contours is SW-NE. In contrast, CWB (Fig. 1b) involves the cyclonic wrapping of low-PV around a high-PV trough, and the orientation of the PV contours is SE-NW. In addition, the centeroid of the CWB is located in the poleward (i.e., cyclonic) side of the upper-level zonal flow. The position and tracks of the low-level cyclones and anticyclones are strongly related to these breaking events, but it is difficult to identify any such relations from isolated examples. A deeper investigation of the relation between RWB events and low-level weather systems is given in Sections 3 and 4.

The frequency distribution of the RWB centroids (Fig. 1c,d) shows that AWB events (Fig. 1c) occur more in the downstream region of the Atlantic ocean basin, maximizing over western Europe, and is sandwiched between the anticyclonic side of the time-mean Atlantic jet and the cyclonic side of the Subtropical time-mean African-Asian jet, while CWB events (Fig. 1d) occur more in the upstream region of the Atlantic ocean, mainly in the cyclonic side of the Atlantic jet. A secondary maximum in CWB is also found poleward of the African-Asia jet. These results are generally in accordance with previous studies (e.g., Strong and Magnusdottir, 2008; Zhang and Wang, 2018).

Note that we have initially tested a RWB detection algorithm similar to the one used in Ndarana and Waugh (2010) and Garfinkel and Waugh (2014), and found qualitatively similar results (see Fig. S1 and Fig. S2 in the SI). However, unexpected extensive CWB activity was detected in the region where AWB is most frequent (Fig. S3), which is why an alternative algorithm was eventually used. Nonetheless, the equivalency of the results using the two different methods gives confidence in our results.

## 2.3 Composites of RWB events

The RWB identification algorithm and the storm-tracking results are used to construct composites of the flow during breaking events in the North-Atlantic, centered either around the centroids of the breakings, or around the corresponding cyclones and anticyclones. The breaking events are first separated into AWB events and CWB events, and composites are then constructed by placing a box sized 60 degrees in latitude by 70 degrees in longitude around the breaking centroid. This is performed separately for all AWB and CWB events, which are then averaged together for each type of breaking. Overall, there are 2,833 AWB events and 2,219 CWB events which satisfy the criterions in Section 2.2, and are used for the breaking-centered composites. Note that the composites are performed on pressure levels (the 850 hPa pressure level is used for the low-level flow, for consistency with the tracking algorithm, and the 250 hPa pressure level is used for the upper-level flow), whereas the RWB events are detected on the 350K isentrope. However, similar results are found if RWB detection is performed on the 250 hPa level, or if the composites are performed on the 350K isentropic level instead.

For the weather system-centered composites, similar criterions are used, but the composites are now centered on the closest cyclone or anticyclone within a 25 degree distance from the breaking centroid. In order to fit the meridional extent of the

composite box (30 degrees to the north and to the south of the composite center), in practice only RWB events or weather systems whose center is between 30N-60N are kept for the compositing. In addition, only weather systems with intensities (in absolute value) larger than $2 \cdot 10^{-5}$ s$^{-1}$ are used for the analysis.

Overall, there are 2,785 cyclones and 2,085 anticyclones identified as closest to the breaking centroid of AWB, and there are 1,690 cyclones and 1,558 anticyclones identified as closest to the breaking centroid of CWB events. For the weather system-centered composites, we further subset the cyclones during AWBs into those residing to the north (N) (1,424 cyclones) or to the south (S) (1,361 cyclones) of the breaking centroid, while for anticyclones during CWBs we use those residing to the North-East (NE) (417 anticyclones). These choices are motivated by the results presented in Fig. 3, and will become clearer later.

## 3  The relation between RWB events and low-level weather systems

### 3.1  Breaking-centered composites of RWB events

We first examine the wave breaking-centered composites of the upper and lower-level flows in the North-Atlantic at the time of maximum breaking. Consistent with previous studies (e.g., Strong and Magnusdottir, 2008; Kunz et al., 2009), and similar to the example cases shown in Fig. 1, the composite of AWB events (Fig. 2a,b) shows a high PV tongue (trough) wrapping anticyclonically around a low-PV (ridge) with a SW-NE orientation (Fig. 2a). The composite of the upper-level zonal flow (Fig. 2b) shows a split jet structure, with a tilted jet in the upstream region, poleward of the AWB center, and a strong decelerated region close to the breaking center. The upstream tilted structure is consistent with the notion that AWB are associated with a poleward shifted jet. However, there is also an additional downstream zonal and more equatorward jet, whose importance has been mentioned in the context of Mediterranean cyclones (e.g., Flaounas et al., 2015, 2022), and is associated with the African-Asian jet. The split jet structure during AWB will be discussed further when investigating the time-evolution composites in Section 4. The composite of CWB events (Fig. 2c) shows a low-PV tongue (ridge) wrapping cyclonically around a high-PV tongue (trough), with a general SE-NW orientation. The composite of the upper-level zonal flow during CWB events shows a more zonal and southward jet, which is also slightly decelerated close to the breaking center (Fig. 2d).

Also shown in Fig. 2 are the composite low-level (850 hPa) vorticity anomaly (black contours in Fig. 2a,c), and SLP anomaly (black contours in Fig. 2b,d). The anomalous SLP composites during AWB are similar to those found by Strong and Magnusdottir (2008) and Kunz et al. (2009), with a negative SLP anomaly generally to the north of a positive SLP anomaly (similar to the positive NAO SLP dipole). Similar results are found for the 850 hPa vorticity composites (Fig. 2a), with a strong negative anticyclonic vorticity extending below and slightly to the east of the upper-level ridge, and a positive cyclonic vorticity to its north-northwest. The vorticity composites of AWB events also show an additional weaker cyclonic signature to the south of the anticyclonic vorticity, not seen in the SLP composites. We note that the RWB detection algorithm we employ here identifies the breaking maximum in a relatively more mature and developed breaking stage (measured by the spatial zonal extent of the breaking tongue), which is why the vorticity anomalies during AWB appear more N-S oriented rather than NW-SE. The initially used RWB detection algorithm (based on Ndarana and Waugh, 2010 and Garfinkel and Waugh, 2014),

which detects breaking at an earlier stage, highlighted more strongly the NW-SE orientation (e.g., see Fig. S1 and Fig. S2). For CWB (Fig. 2b) we find, consistent with Strong and Magnusdottir (2008) and Kunz et al. (2009), a positive SLP anomaly to the N-NE of a negative SLP anomaly (generally similar to the negative NAO SLP dipole). The vorticity composite during CWB (Fig. 2c) shows a strong cyclonic anomaly extending below and slightly to the east of the upper-level trough, and an anticyclonic anomaly to its NE.

A priori, it is not clear whether these SLP anomalies are mostly signatures of large-scale, slowly varying flow (i.e., signatures of the low-frequency weather regimes), or whether they are associated with the synoptic-scale weather systems (i.e., high-frequency eddies). For example, it has been suggested that the positive and negative polarities of the NAO (which are low frequency modes) are directly linked to RWB events, or more generally to changes in their frequency and location (Benedict et al., 2004; Franzke et al., 2004; Rivière and Orlanski, 2007; Woollings et al., 2008; Strong and Magnusdottir, 2008; Kunz et al., 2009). Here we utilize the storm-tracking algorithm to show a clear relation between RWB events and migrating low-level cyclones and anticyclones, and to investigate the characteristics of low-level weather systems during RWB events. How these feed back into the slowly varying atmospheric modes is left for further study.

### 3.2 Low-level weather system characteristics during RWB events

The relation between upper-level RWB and low-level weather systems is first investigated by examining the relative positions, intensities, and propagation characteristics of the weather systems during RWB events. In the following analysis (presented in Fig. 3-Fig. 5) we use all the cyclones and anticyclones identified within a $25°$ distance from the breaking centroid, and not just the closest features, as done for the composite analysis. Overall, there are 5,106 cyclones and 4,136 anticyclones during AWBs, and 4,216 cyclones and 3,681 anticyclones during CWBs. For AWBs, in 90% of the cases there is at least one anticyclone present in its vicinity, and in 94% of the cases there is at least one cyclone present. For CWBs, these numbers are 92% and 95% for anticyclones and cyclones, respectively. In both types of breaking events, it is very rare (less than 1%) that the upper level RWB event occurs without any surface weather system present in its vicinity.

Fig. 3 shows scatter plots of the positions of cyclones (left column) and anticyclones (right column) during RWB events, where color denotes the intensity of the system (absolute value, in units of $10^{-5}\text{s}^{-1}$), relative to the center of the AWB (Fig. 3a,b) and CWB (Fig. 3c,d). Clear signatures of preferred relative positions arise in these aggregated scatter plots (see black contours denoting the corresponding PDFs. During AWB events (Fig. 3a,b), anticyclone locations are mostly within the upper-level ridge, close and slightly to the north of the AWB center (denoted by the cross symbol), while two distinct locations emerge for cyclones; intense cyclones are typically found to the N-NW of the breaking center, and a secondary region of weaker cyclones is found to the S-SE of the breaking center. Hence, AWBs are often associated with a cyclone-anticyclone-cyclone (C-AC-C) triple, consistent with the vorticity composites shown in Fig. 2a. Analyzing how often this tripole occurs simultaneously reveals that in 56% of the cases a C-AC-C structure that is N-S oriented is observed (i.e., at least one anticyclone, with at least one cyclone to its north and one cyclone to its south). During CWB events, cyclones are typically found at low-levels close to the trough region, slightly to the SW of the CWB center (Fig. 3c). For anticyclones during CWB, the relative positions are more spread, but slightly more intense anticyclones are found to the E-NE of the breaking center

(Fig. 3d). Note that for the initially used RWB detection algorithm (based on Ndarana and Waugh, 2010), the signature of intense anticyclones to the NE of CWBs was much clearer (see Fig. S2d in the SI).

The geographical distribution of the weather systems and their propagation characteristics are important for regional weather (for example in the European-Asian continent and Mediterranean region), and can influence the distribution of, e.g., precipitation and extremes. Since AWB and CWB events occur most frequently in different regions over the Euro-Atlantic region (Fig. 1c,d), and since the relative distribution of the weather systems is different in each one of the cases (Fig. 3), it is also of interest to examine where cyclones and anticyclone reside, in physical space, during AWB and CWB events. The thick black contours in Fig. 4 show PDFs of the weather system counts (calculated using a kernel density estimator and multiplied by the number of systems), highlighting the locations where they are most observed. Motivated by the distinct relative positions found in Fig. 3a for cyclones during AWB, in Fig. 4a the PDFs are further separated into cyclones residing to the north (black thick contours) and south (gray thick contours) of the breaking center.

Cyclones during AWB events (Fig. 4a) are generally more spread over the Atlantic ocean basin and the downstream region of the storm-track, with more cyclones reaching the UK and Scandinavia, compared to cyclones during CWB (Fig. 4c), which are more concentrated in the western side of the ocean basin, roughly co-located with the region where CWB events are most frequent. The secondary peak of cyclones residing to the south of the breaking center (gray thick contours in Fig. 4a) is found mostly in the downstream subtropical Atlantic and in the Mediterranean region. Similarly, anticyclones during AWB events (Fig. 4b) are found more frequently in the downstream region of the Atlantic ocean basin (close to the region where AWB events are most frequent), while anticyclones during CWB (Fig. 4d) are found more frequently in the mid and upstream region. The latter also exhibit a secondary local maximum over Greenland.

The weather system characteristics are examined more quantitatively by examining the histogram distributions of their intensity (absolute value), spatial position, and longitudinal and latitudinal track displacements during the two types of RWB events (Fig. 5). Cyclones to the south of AWB are much weaker compared to cyclones to the north of the AWB center and also compared to cyclones during CWB events (Fig. 5a), while anticyclone intensities are similar between the two types of breakings (Fig. 5f). There is a clear longitudinal separation between the positions of the weather systems, with both cyclones and anticyclones being located more upstream during CWB compared to those during AWB (Fig. 5b,g). The latitudinal separation of cyclones during AWB is by construction, but it can also be seen that cyclones during AWB can be found at much lower latitudes compared to cyclones during CWB (thick and thin black lines in Fig. 5c, respectively). The latitudinal distribution of anticyclones during CWB and AWB is similar, but anticyclones during CWB are slightly more concentrated to the north (Fig. 5h, difference is statistically significant at the 5% level).

The weather systems also have very distinct propagation characteristics, depending on the type of breaking (cyclonic/anticyclonic), and their position relative to the breaking center. Fig. 5d,e and Fig. 5i,j show the longitudinal and latitudinal displacements of cyclones during the five days centered on the breaking maximum (i.e., the difference between the position two days after the breaking maximum, $r_{b+2}$, minus the position two days prior the breaking maximum, $r_{b-2}$; $\Delta r = r_{b+2} - r_{b-2}$). Cyclones during CWB (thin black line in Fig. 5d,e) mostly propagate eastward and poleward, as expected (e.g., Gilet et al., 2009; Rivière et al., 2012; Tamarin and Kaspi, 2016). However, separating cyclones during AWB into those occurring to the N

and to the S of the AWB center shows that while cyclones to the N (dashed red lines in Fig. 5d,e) also tend to move eastward and poleward, cyclones to the S (dash-dotted blue lines in Fig. 5d,e) are much more stationary zonally and meridionally, and even propagate on average slightly equatorward. The propagation characteristics associated with this subset of cyclones (to the S of AWBs), which is expected given the anticyclonic (southwestward) wrapping of the trough around the ridge, could be investigated further for Mediterranean cyclones, for which AWB was suggested to play a crucial role (Flaounas et al., 2015; Raveh-Rubin and Flaounas, 2017; Flaounas et al., 2022). Finally, anticyclones during AWB and CWB events have similar meridional propagation characteristics (Fig. 5j, difference is not statistically significant), while the eastward displacements of anticyclones during AWB are slightly larger compared to anticyclones during CWB ((Fig. 5i, an averaged longitudinal displacement of 31.2° compared to 27.8°, respectively, difference is statistically significant at the 5% level).

## 4    The life-cycles of cyclones and anticyclones in the North Atlantic

We first examine composites of cyclones and anticyclones in the North Atlantic during the time of maximum intensity, overlaid with the coincident RWB PDF frequencies (estimated as Kernel Density Estimators and multiplied by the number of events) (Fig. 6). We also quantify the percentage of cyclones or anticyclones that are associated with RWB in their vicinity (defined as less than 25 degrees to their north/south or east/west) sometime during their time evolution. We find that 69% of the cyclones are associated with an AWB in their vicinity, 67% with a CWB in their vicinity (with 47% having both), while 11% do not have a RWB occurring in their vicinity during their lifetime. For anticyclones, 65% are associated with an AWB in their vicinity, 61% with a CWB in their vicinity (with 41% having both), while 15% do not have a RWB occurring in their vicinity during their lifetime. These percentages do not change much when taking only the 50 strongest weather systems from each season.

The results imply that most cyclones and anticyclones in the North Atlantic are involved with breaking at some point during their lifetime. Interestingly, slightly more cyclones are associated with AWB rather than CWB. This is consistent with the fact that AWBs are generally more frequent, but is also probably related to diabatic heating associated with the warm conveyer belt, which contributes to the upper-level ridge development (e.g., Grams et al., 2011; Pfahl et al., 2015; Methven, 2015). The AWB occurs mostly to the south-east of the cyclones, but the AWB relative positions are rather spread around the cyclone such that overall, the AWB frequency around cyclones is low (Fig. 6a). On the other hand, for cyclones associated with CWB, the breaking occurs in a similar position (close to their center and slightly to the north-east), hence the CWB frequency is high, even though there are generally fewer CWB events compared to AWB events (Fig. 6c). A similar but opposite picture is found for anticyclones: for those associated with an AWB the breaking typically occurs in a similar relative position (close to their center and slightly to the south), hence the AWB frequency PDFs are high (Fig. 6b), while the CWB positions are more spread around the anticyclones, hence the frequency PDFs of CWB are lower (Fig. 6d). These results are consistent with our earlier findings presented in Fig. 3 (from the perspective of the breaking center).

Fig. 6 shows that compositing over all cyclones and anticyclones in fact mixes between cyclonically and anticyclonically breaking systems. This motivates our further decomposition of cyclones and anticyclones into those breaking cyclonically and anticyclonically, to examine their distinct time evolution and characteristics. We denote this classification as "same-sense"

weather system vorticity and RWB type (e.g., anticyclones with an AWB, and cyclones with a CWB), and "opposite-sense" weather system vorticity and RWB type (e.g., anticyclones with a CWB, and cyclones with an AWB). The time evolution of the weather systems during RWB events is investigated by performing composites relative to the center of the closest weather system for each type of RWB (as described in the Methods, see Section 2).

## 4.1 Same-sense weather system vorticity and RWB type

We begin with an examination of the cases with same-sense weather system vorticity and RWB type. Fig. 7a-e and Fig. 9a-e show composites of the upper-level PV anomaly and the low-level SLP anomaly (in contours), centered on the anticyclones for AWB and on the cyclones during CWB, from $T = -3$ days prior to the breaking and up to $T = 2$ days after the breaking. During the time of maximum breaking ($T = 0$, Fig. 7c and Fig. 9c), the composites show a structure similar to that obtained by centering the flow on the corresponding wave breaking type (shown in Fig. 2). The signal of the upper-level breaking (e.g., characterized by the overturning of the PV contours) is slightly weaker in the weather system-centered composites. However, the results are otherwise similar, and capture the different cyclone/anticyclone orientations found for AWB and CWB events. The same-sense composites also give results similar to those obtained by compositing the flow on all the systems, regardless of whether an upper-level RWB event has occurred (Fig. 6). This is because the centers of anticyclones and AWB events are more co-located (and similarly for cyclones and CWB events), and hence they dominate the composites. Since the same-sense weather system vorticity and RWB cases are more representative of the composite life-cycle of anticyclones and cyclones in the North Atlantic, we explore these cases in more detail, but in Section 4.2 we also compare these cases to the opposite-sense weather system vorticity and RWB cases.

Note that the compositing procedure will inevitably highlight the intensity of the composited feature, and this should be taken into account wherever claims are made which are based on the intensity of the system in the composite. For example, compositing on anticyclones during AWB will result with a strong low-level anticyclone and an anomalous anticyclonic circulation at upper levels. Nonetheless, compositing the flow on cyclones during AWB instead still gives a strong anticyclonic circulation at upper-levels, even though the compositing is on the cyclones in this case. In addition, it is also of interest to compare between composites centered on anticyclones during AWB relative to those during CWB. While in both cases the centering is on the anticyclones, the composites reveal quite distinct life-cycles, as will be shown (and similarly for cyclones during AWB or CWB events).

### 4.1.1 Anticyclones during AWB

The time evolution of composites centered on the anticyclones during AWB (Fig. 7a-e) reveals some interesting features. Three days prior to the breaking ($T = -3$ days, Fig. 7a), there is a strong low-level anticyclone residing to the east of an upper-level ridge, and a low-level cyclone initially to the west of the anticyclone. Both the ridge and the anticyclone reside in the anticyclonic side of the upper-level jet (Fig. 7f), as well as in the anticyclonic side of the time-averaged mean jet (this is more visible at $T = -1$, see Fig. S4a,b in the SI).

During the build-up of the wave breaking, the anticyclone and the cyclone to its west slightly intensify (Fig. 7a,b) and the cyclone then rotates in an anticyclonic manner relative to the anticyclone (Fig. 7a-e), eventually merging with a negative SLP anomaly initially to the northeast of the anticyclone. Parallel to this, the upper-level trough to the east (downstream) of the ridge is wrapped around the ridge and the classical picture of wave breaking and inversion of meridional PV gradient is found (i.e., a negative PV anomaly to the north of a positive PV anomaly). During the decay stage of the wave breaking (Fig. 7d,e), the anticyclone slightly weakens and the flow becomes more barotropic, and two days after the breaking maximum ($T = 2$ days, Fig. 7f), the low pressure anomaly is entirely to the north of the anticyclone. The low-level positive NAO-like pressure dipole (the low-above-high pressure anomalies) is in agreement with Strong and Magnusdottir (2008) and Kunz et al. (2009), but here the time-evolution of the low-level cyclones and anticyclones leading to this structure is examined.

The total upper-level zonal flow weakens significantly to the north of the ridge during the evolution of the breaking, while the downstream zonal and more southward jet intensifies (Fig. 7f-j). Hence, the initially more wavy-like upper-level jet (Fig. 7f) is split into an upstream tilted jet, and a downstream zonal jet (Fig. 7h). The split jet structure has been noted in several previous studies examining Mediterranean cyclones and their relation to AWB (e.g., Flaounas et al., 2015; Raveh-Rubin and Flaounas, 2017). However, it is not usually discussed in relation to the classical theories concerning the poleward shift of the jet in AWB. Separating the total flow into a time-mean (in this case the climatological DJF mean over all the years) and an anomalous flow, i.e., $u = \overline{u} + u'$ (where an overline represents time-mean and prime represents deviation from that time-mean) shows that it is mainly the time-mean flow that contributes to the downstream jet (note that the composites of the time-mean flow vary with time here since the compositing is centered on the weather systems which are moving, hence at different locations with respect to the Eulerian time-mean flow). The downstream jet is therefore probably linked to the time-mean subtropical Asian-African jet, as the weather systems approach the eastern side of the ocean basin.

The negative upper-level meridional wind between the ridge and the trough to its east (Fig. 7k-o) increases significantly during the breaking, consistent with the intensifying anomalous ridge-trough system. Similarly, the negative anomalous zonal wind between the ridge and the trough which is breaking anticyclonically to its south is also intensifying (see also Fig. S4f-o in the SI for the anomalous velocities). The anomalous southwestward upper-level wind is contributing to the breaking of the wave by advecting the anomalous upper-level trough southward and westward around the ridge. This nonlinear advection reinforces the upper-level anticyclonic rotation due to the background anticyclonic shear. In addition, the anomalous upper-level anticyclonic circulation associated with the ridge induces a relative anticyclonic rotation at low-levels (through interaction between the upper and lower levels PV anomalies, not shown), as it acts to advect the low-level anticyclone equatorward and the cyclone to its west poleward (see SLP anomalies and arrows representing the anomalous circulation in Fig. 7a-e), in general agreement with Gilet et al. (2009); Rivière et al. (2012); Tamarin and Kaspi (2016).

Fig. 8a-c shows cross sections of the upper-level zonal wind at the longitude crossing the center of the composite box (i.e., the center of the anticyclone). The peak of the total upper-level zonal flow initially increases slightly and then decreases, while the relative latitude at which this peak is achieved remains roughly the same (Fig. 8a, see changes in the lines going from black, denoting $T = -3$ days, to blue, denoting $T = 2$ days). Southward of the peak in total $U$, the zonal flow decreases significantly during the breaking, and a secondary peak develops further southward. The strong deceleration of $U$ southward of the anti-

cyclone center is related to the intensification of the negative anomalous zonal wind (Fig. 8b), due to the intensifying ridge
and the anticyclonically wrapping trough to its south. The climatological (time-averaged) zonal wind (Fig. 8c) also contributes
to the apparent weakening of $U$, due to the motion of the weather systems into a region where the time-mean Atlantic jet is
weaker, while it contributes to the strengthening of the total $U$ more southward (a signature of the downstream subtropical
jet). These changes in the time-mean jet are related to the eastward propagation of the anticyclones, as they propagate away
from the Atlantic jet and approach the downstream exit region of the storm-track. Note that the anticyclones are also propa-
gating meridionally. Taking into account the averaged latitudinal displacement of the anticyclones at each time-step (which is
435 poleward in this case, see Fig. 5d), shows that the peak in the total upper-level $U$ is shifted poleward (see Fig. S6a in the SI).

The results above are consistent with the usual notion that AWB events are associated with a poleward shift of the zonal mean
jet, due to the poleward momentum fluxes that result from the SW-NE tilt of the PV contours. Here we suggest a mechanistic
interpretation, in which the poleward shift is a result of the intensification and anticyclonic rotation of the ridge-trough system.
The anomalous velocities associated with an isolated intensifying ridge would contribute to a local poleward shift of the jet (a
440 strengthening of the total zonal flow poleward the ridge, and a weakening equatorward of it), similar to what is observed during
anticyclonic blocking events (Tyrlis and Hoskins, 2008; Woollings et al., 2008). However, for a linear wave, this effect will
cancel out (in the zonal mean) with the adjacent trough, which will have the opposite net effect. The nonlinearity associated
with the breaking results in a relative northeastward motion of the ridge and a relative southwestward motion of the downstream
trough (i.e., a relative anticyclonic rotation). This breaks both the zonal and the meridional symmetries. An asymmetry in the
445 meridional velocity forms since it is mainly the negative meridional wind between the ridge and the developing downstream
trough that intensifies (as opposed to the positive meridional velocity between the ridge and the upstream trough). Similarly,
a negative anomalous zonal wind forms between the ridge and the trough which is wrapping to its south. These anomalous
velocities are not averaged out in the zonal mean in this case. Thus, an intensifying and anticyclonically breaking ridge-trough
system results in an intensification of the total zonal flow in the area northward of the breaking and a weakening southward of
450 the breaking.

Overall, for anticyclones during AWB events, we see a downstream trough development, equatorward of the upper-level jet,
and an anticyclonic relative rotation at both upper and lower-levels. Hence, in this case, the breaking and relative rotation are
in the same sense as the weather system circulation (i.e., anticyclonic), which is acting to reinforce it (see also the schematic
shown in Fig. 11a for an illustration of the upper and lower level evolution for composite anticyclones during AWB, discussed
in the conclusions).

### 4.1.2 Cyclones during CWB

We next examine the time evolution of composites centered on cyclones during CWB (Fig. 9a-e). Three days prior to the
breaking ($T = -3$ days, Fig. 9a), there is a strong low-level cyclone, residing to the east of an upper-level trough, and poleward
of an upper-level jet (i.e. in the cyclonic side, Fig. 9f). There are also weak signatures of an anticyclone to the west and to the
460 north-northeast of the cyclone. During the build-up of the breaking (Fig. 9a-c), both the low-level cyclone and the upper-level
trough intensify, while a downstream ridge and a low-level anticyclonic anomaly are intensifying to the east and north-east of

the cyclone. A relative cyclonic rotation is observed both at upper and lower-levels, as the downstream ridge starts rotating in a cyclonic manner relative to the trough (Fig. 9b-d).

During the decay stage of the wave breaking (Fig. 9d,e) the ridge is weakened and dissipated out (see also black contours in Fig. 9n,o showing the PV anomaly). At low-levels, the cyclone becomes more barotropically aligned with the upper-level trough, and two days after the breaking maximum ($T = 2$ days, Fig. 9e) the high pressure anomaly is mostly to the north of the cyclone. These results are generally similar to Strong and Magnusdottir (2008), who examined composites of CWB and found a low-level negative NAO-like pressure dipole (a high-above-low pressure dipole), and to Kunz et al. (2009), who examined CWB in simplified GCM experiments and found a similar dipole at upper-levels (a low-above-high meridional PV anomaly dipole).

The total zonal flow (Fig. 9f-j) weakens in magnitude, most strongly to the north of the cyclone center, and its peak shifts southward relative to the cyclone center. The weakening and southward shift of the jet in the composites, which are observed also in the composites of the time-mean flow (Fig. S5a-e, see also Fig. 8d-f), are probably a result of the eastward and poleward motion of the cyclones, as they move further away from the time-mean flow and approach the weaker jet exit region. The anomalous upper-level zonal wind (arrows in Fig. 9a-e), which is dominated by the upper-level trough, is positive to the south of the trough, but a strong negative anomalous zonal wind is generated between the trough and the ridge which is breaking cyclonically to its north. Similarly, the positive upper-level meridional wind between the trough and the ridge to its east, which is mostly due to the anomalous flow (see also Fig. S5f-o for the anomalous winds), increases during the evolution of the breaking, as the trough-ridge system grows (Fig. 9k-o). As in AWB, the anomalous upper-level wind is contributing to the breaking, in this case by nonlinear northwestward advection of the ridge around the trough (i.e., in a cyclonic manner), which reinforces the rotation induced by the cyclonic shear. Hence, the anomalous upper-level wind is contributing to both the downstream development of the ridge (through linear advection), and to the cyclonic rotation (through nonlinear advection). Note that the downstream ridge development in this case is also related to the indirect influence of diabatic heating. Latent heat release associated with the warm conveyer belt of the cyclone contributes to a negative PV tendency aloft. The negative PV tendency is contributing to the ridge development through an upward negative PV advection (not shown, consistent with e.g., Grams et al., 2011; Pfahl et al., 2015; Methven, 2015).

The cross sections of the upper-level zonal wind at the longitude crossing the center of the cyclone (Fig. 8d-f) show that the peak of the total zonal flow (Fig. 8d) weakens and shifts southward throughout the life-cycle of the breaking. This is mainly due to the time-mean zonal flow (Fig. 8f), and is related to the eastward and poleward propagation of the cyclones away for the jet core. In addition, the flow weakens strongly poleward of the breaking. The latter is due to the intensification of the negative anomalous zonal wind (Fig. 8e), related to the intensifying and cyclonically rotating ridge-trough system. Adjusting the cross-sections by taking into account the averaged latitudinal poleward displacement of the cyclones at each time-step shows that in this case, the peak in the total upper-level $U$ remains roughly at the same latitude (see Fig. S6b in the SI). This seems to suggest an interesting asymmetry between AWB and CWB events, namely that during AWB events, the already poleward shifted jet can shift even further poleward, while during CWB, the equatorward shifted jet remains at a similar latitude.

The weakening of the total upper-level jet poleward of the breaking during CWB is consistent with the notion that CWBs are associated with an equatorward shifted zonal mean jet, due to the equatorward momentum fluxes that result from the SE-NW tilt. Similar to AWB, we suggest that this is due to nonlinearity associated with the breaking (i.e., the cyclonic wrapping of the ridge around the trough) which breaks the zonal and meridional symmetries, as it favors a positive meridional wind between the trough and the ridge to its east, and a negative anomalous zonal wind between the trough and the breaking ridge to its north, which are not averaged out in the zonal mean.

Overall, for cyclones during CWB events, we see a downstream ridge development, poleward of the upper-level jet, and a relative cyclonic rotation at both upper and lower-levels. Hence, in this case too, the breaking and relative rotation are in the same sense as the weather system circulation (see also Fig. 11c for a schematic illustration of the upper and lower level evolution for composite cyclones during CWB). Such a development is similar to what is usually observed or expected for cyclones, which reside on average in the cyclonic side of the jet, and experience a cyclonic wrap-up at upper-levels (e.g., Figure 4 in Dacre et al., 2012).

## 4.2 Opposite-sense weather system vorticity and RWB type

We next investigate the life-cycle of weather systems during RWB which is occurring in the opposite-sense to their rotational flow. For this we examine composites centered on anticyclones during CWB (to the NE of the CWB center), and on cyclones during AWB events, where the latter is separated into cyclones residing to the N or S of the AWB center. We note that similar results are found if the separation is into cyclones residing to the NW or SE of the AWB center, (not shown).

### 4.2.1 Anticyclones during CWB

Compositing the flow on anticyclones to the NE of CWBs (Fig. 10a-e) shows that they are located now to the north of the zonal jet (i.e., in the cyclonic shear), even two days prior to the breaking (Fig. 10a). These composites are dominated by an upper-level ridge, and a low-level anticyclone with a weaker cyclone to its SW (similar to the orientation found in the CWB composites). Although the anomalous upper-level wind is dominated by the anticyclonic circulation, the cyclone and anticyclone still rotate in a cyclonic manner relative to each other (now it is the cyclone that rotates cyclonically around the anticyclone, due to the centering of the composites around the anticyclones). Similarly, the deepening trough seen to the west of the anomalous ridge (see the 4.5 PVU contour) is clearly not a result of advection associated with the anomalous upper-level anticyclonic circulation, which is in the opposite sense. One day after the breaking maximum (Fig. 10e), signatures of AWB start to develop in the downstream region of the ridge.

Overall, in the case of anticyclone composites during CWB, we find that the breaking and relative rotation are occurring in an opposite sense to the anomalous anticyclonic circulation associated with the anticyclone (see also the schematic shown in Fig. 11b for an illustration of the upper and lower level evolution for composite anticyclones to the NE of CWB). This subset includes anticyclones developing from upstream cyclones that involve CWB. A well studied example of such cases are blocking anticyclones which are preceded by explosive cyclones (e.g., Colucci, 1985; Lupo and Smith, 1995). Note, however,

that from the longitudinal and latitudinal displacement PDFs of these anticyclones (Fig. 5i,j), stationary blockings constitute only a small subset of these features.

### 4.2.2  Cyclones during AWB

For AWB, we separate the composites into those centered on cyclones to the north (N) (Fig. 10f-j) and to the south (S) (Fig. 10k-o). In both cases, in addition to the cyclonic upper-level circulation associated with the anomalous trough, there is also a strong anomalous upper-level ridge, even though the compositing is performed on the cyclones.

For cyclones to the N (Fig. 10f-j), prior to the breaking there is a strong low-level cyclone and an anticyclone to its SE, with the former residing in the poleward (cyclonic) side of the jet, and the latter residing on the equatorward (anticyclonic) side of the jet. There is also another weaker anticyclone initially to the west (upstream) of the cyclone. The upstream anticyclone rotates slightly cyclonically relative to the cyclone, similar to what was found for cyclone composites during CWB. However, the downstream anticyclone rotates in an anticyclonic manner relative to the cyclone. This anticyclonic rotation is related to the dominance of the anomalous ridge at upper-levels, and is in stark contrast to the low-level cyclonic circulation and the usual cyclonic wrap-up seen at upper-levels during the life-cycle of cyclones (e.g., Figure 4 in Dacre et al., 2012). The strong upper-level ridge ultimately leads to an anticyclonic breaking in the downstream region, by contributing to the growth and anticyclonic breaking of the trough to the east of the ridge. Note that the breaking signal is very weak due to the centering of the flow over the cyclones, which are not necessarily close to the breaking center.

Overall, for cyclones residing to the N of AWB events, we see a relative anticyclonic rotation at both upper and lower-levels, hence the breaking and relative rotation are in the opposite sense to the weather system circulation. However, despite the composite being centered on the cyclone, the anomalous upper-level circulation has a very clear anticyclonic ridge, which enhances the anticyclonic wave breaking and rotation. In this respect, the anomalous circulation is not in an opposite-sense to the wave breaking. This subset of cyclones likely includes the "upstream cyclones" discussed in the context of blocking events (e.g., Colucci, 1985; Lupo and Smith, 1995), which often involve AWB (e.g., during the Scandinavian blocking onset studied in Michel et al., 2012). It probably also includes cases where the cyclone is in the anticyclonic side of the upper-level jet, which is deflected poleward during the AWB (i.e., cyclones within the ridge area). An example of such a cyclone was presented as a proposed analog to the anticyclonic barotropic shear (LC3) idealized frontal-wave cyclone (Shapiro et al., 1999). This complicates the classification of this subset of opposite-sense cyclones, since it includes a few different cyclone evolutions. We therefore did not include a schematic illustration of this case in Fig. 11.

Finally, centering the flow on cyclones to the S of the AWB center (Fig. 10k-o) gives, prior to the breaking, a strong anomalous upper-level trough and a weaker anomalous upper-level ridge to its NW. At low levels, there is an anticyclone initially to the W-NW of the cyclone. The anticyclone is relatively strong, given that the composites are centered on the cyclones. The low-level anticyclone-cyclone dipole as well as the upper-level ridge-trough dipole then rotate in an anticyclonic manner relative to each other, with the anticyclone becoming ultimately to the north of the cyclone. Hence, these cyclones exhibit a composite time evolution that is quite different from both cyclones during CWBs, and from cyclones residing to the N of AWB events. While the former two exhibit a downstream ridge development (with or without a cyclonic wrap-up,

respectively), the composite evolution of cyclones to the S of AWBs exhibit an upstream ridge development and an anticyclonic relative rotation. In addition, it can be seen that the cyclonic anomalous winds in this case (arrows in Fig. 10k-o) do not contribute to the upstream ridge development and the overall relative anticyclonic rotation.

Overall, for cyclones to the S of AWB events, we see an upstream ridge development, and a relative anticyclonic rotation at both upper and lower-levels. Hence, the breaking and relative rotation are in an opposite sense to the weather system circulation (see also Fig. 11d for a schematic illustration of the upper and lower level evolution for composite cyclones to the S of AWB). This subset of opposite-sense cyclones includes subtropical and tropical cyclones (Davis, 2010; Galarneau et al., 2015; Bentley et al., 2017) and Mediterranean cyclones (Flaounas et al., 2015; Raveh-Rubin and Flaounas, 2017; Flaounas et al.,

2022) forming due to PV streamers associated with AWBs (see Fig. 4a). The similarity between the weather system-centered composites for cyclones to the S of AWB (Fig. 10k-o) and Fig. 10 of Flaounas et al. (2015) showing weather system-centered composites for the 200 most intense Mediterranean cyclones is remarkable, which further highlights the importance of Atlantic AWB events in the development of Mediterranean cyclones.

The opposite-sense composites investigated here show very different cyclone and anticyclone time evolutions compared to

the same-sense composites presented in Section 4.1. These different life-cycles are often missed when performing composites over all systems.

## 5    Conclusions

Applying automated detection algorithms of upper-level RWBs and low-level weather systems has allowed us to examine in a more systematic and comprehensive way the relation between weather systems and RWB events in the North Atlantic, and

hence to complement and generalize idealized wave life-cycle experiments (Simmons and Hoskins, 1978; Davies et al., 1991; Thorncroft et al., 1993) and single-event case studies (e.g., Shapiro and Granas, 1999). Going back to the first two questions posed in the introduction, the main results can be summarized as follows:

Composites of cyclones and anticyclones involve a mixture of different types of life-cycles, depending on whether they involve CWB or AWB, as well as their position relative to the RWB center. Moreover, weather system characteristics (including

actual and relative positions, intensities, and displacements) differ depending on the associated breaking type. We find that in the North Atlantic, most cyclones and anticyclones are associated with an AWB and/or CWB at some point during their lifetime (more than 60%), with a large portion having both wave breaking types (more than 40%), and few weather systems having neither (less than 15%, see first paragraph of section 4). AWB is generally more frequent than CWB, hence slightly more weather systems are found with AWB. We also find that the centers of CWBs during cyclones and AWBs during anticyclones

are spatially co-located near the respective surface weather system center.

During AWB, a low-level anticyclone is found close and slightly to the N of the breaking center. However, two preferred locations of cyclones relative to the center of the breaking emerge, to the S-SE and to the N-NW of the anticyclone (Fig. 3a,b). Overall, the orientation of the cyclone-anticyclone-cyclone tripole is SE-NW during the breaking development stage, and becomes more S-N oriented by the end of the life-cycle. Geographically, cyclones to the N-NW of AWB are spread more over

the upstream and mid-Atlantic ocean basin, while cyclones to the S-SE (which are generally much weaker) are found more over the subtropical eastern Atlantic and the Mediterranean region (Fig. 4a,b). The propagation characteristics of these two groups of cyclones differ significantly; while the cyclones to the N-NW propagate on average eastward and poleward, cyclones to the S-SE of AWB propagate much less zonally and meridionally.

During CWB, a strong low-level cyclone is usually found close to the CWB center, while anticyclones are found mainly to the NE of the cyclones, such that the cyclone-anticyclone dipole has a SW-NE orientation (Fig. 3c,d). Geographically, cyclones during CWB are found much more westward (mostly close to the western coast of the North Atlantic ocean basin) and at slightly higher latitudes compared to cyclones during AWBs. Anticyclones during CWBs are also located more upstream compared to anticyclones during AWBs (Fig. 4c,d).

Given the different weather system characteristics summarized above, we distinguish between "same-sense" cases (i.e., cyclones during CWB, and anticyclones during AWB) and "opposite-sense" cases (i.e., cyclones during AWB, and anticyclones during CWB). Compositing the cyclones and anticyclones based on this criterion, we find that in similar pairings the surface system is positioned so that its associated upper-level winds would enhance the breaking which is in the same sense, but for opposite pairings, the upper-level winds associated with the surface system do not act to enhance the breaking which occurs in the opposite sense. Correspondingly, the same-sense and opposite-sense composites show very different life-cycles and time evolutions, depending on the type of upper-level RWB they are associated with, as well as the position of the surface cyclone relative to the breaking in the AWB case. The different evolutions are shown schematically in Fig. 11.

For anticyclones, we find that in both types of life-cycles, the anticyclone is initially to the east of the cyclone, and the anomalous upper-level velocity is dominated by an anticyclonic circulation. However, there are major differences between the two cases. First, the anticyclone and the upper-level ridge are in the anticyclonic side of the jet in AWB, and in the cyclonic side of the jet in CWB. Moreover, it is the downstream trough that deepens in the AWB composites, while it is the upstream one in the CWB composites. Lastly, the relative rotation is anticyclonic during AWBs, while it is cyclonic during CWBs. The two types of life-cycles are shown schematically in Fig. 11a and Fig. 11b, respectively. For cyclones, comparing the cyclones during CWB to cyclones to the S of AWB, we find that in both cases the cyclone is initially to the east of the anticyclone, and the anomalous upper-level velocity is dominated by a cyclonic circulation. However, we find a downstream ridge development and cyclonic relative rotation in the CWB case, and an upstream ridge development and an anticyclonic relative rotation in the AWB case. The two types of life-cycles are shown schematically in Fig. 11c and Fig. 11d, respectively.

To address the third question posed in the introduction, we compare our results with the idealized life-cycles studied in the literature. We note that in our RWB decomposition, AWB includes both LC1 and P2 type breaking events, while CWB includes both LC2 and P1 type breaking events. We find that the life-cycles of cyclones and anticyclones during AWB are similar to the idealized nonlinear normal-mode development for the anticyclonic shear case of Davies et al. (1991) (see their Fig.8) and the later stages of the LC1 wave life-cycle of Thorncroft et al. (1993) (see their Fig.6, day 6 and onwards). In all cases, the dominant motion is a relative anticyclonic rotation of the upstream cyclone and downstream anticyclone. For the CWB cases, in our results the dominant evolution is a poleward motion of the anticyclone to the east (downstream) of the cyclone, which is wrapping cyclonically around the cyclone. This is similar to the idealized nonlinear normal-mode development for the cyclonic

shear case of Davies et al. (1991) (see their Fig.9), but is different from the LC2 life-cycle of Thorncroft et al. (1993), in which the dominant evolution is an equatorward motion of the anticyclones to the west (upstream) of the cyclone (see their Fig.9). In all cases, however, the relative rotation between the cyclone and the anticyclone is cyclonic. Note that the general similarity to the idealized life-cycles of Davies et al. (1991) and Thorncroft et al. (1993) is not a priori obvious, given that in the idealized simulations the initial perturbation is a pure zonal wave and the breaking type is controlled artificially (e.g., by changing the

background meridional shear). It is consistent, however, with the fact that the observed RWB is typically found where the background meridional shear is of similar sense.

Our analysis helps put together the results obtained from previous idealized life-cycle studies and observational case-studies to obtain a more coherent picture of the relation between RWB and surface weather systems in the North Atlantic. Note that the methods used in the current study can be easily applied to other regions, e.g., the Pacific storm-track or the Southern

Hemisphere storm-track, where we also expect to find different surface-system and RWB configurations. The classification into different sub-classes of cyclones or anticyclones based on configurations of weather system types and RWB types and their corresponding life-cycles may have a few potential implications.

Given that the time evolutions of the different subsets involve different wave-mean flow interactions and jet shifts, and different weather system characteristics such as intensities, positions and displacements, correctly identifying them may help

improve weather prediction of subsequent development. For example, the upstream cyclones found to the W of anticyclones during both AWBs and CWBs may fall under the upstream cyclone theory suggested for block onset mechanism (Colucci, 1985; Lupo and Smith, 1995). Maddison et al. (2019) examined the role of the upstream cyclone on the predictability of block onsets over the Euro-Atlantic Region. They showed that block onset in the case studies is sensitive to changes in the forecast location and intensity of upstream cyclones in the days preceding the onset, and concluded that improvement in the forecasts of

the upstream cyclone may help improve block onset forecasts. The upstream cyclone and developing downstream block may involve either a CWB or an AWB (or both, for an omega-type block). The subset of cyclones to the S of AWB may be relevant to studies which discuss the influence of antecedent AWB events on subsequent tropical, subtropical, or Mediterranean cyclone development (e.g., Appenzeller et al., 1996; Davis, 2010; Galarneau et al., 2015; Flaounas et al., 2015; Bentley et al., 2017; Portmann et al., 2021; Flaounas et al., 2022).

It would be interesting to investigate if similar results are obtained, e.g., in a zonally symmetric storm-track, or whether some of these findings are shaped by the stationary waves in the North Atlantic. Another interesting future direction is to incorporate circulation regimes in the North Atlantic, which are recurrent and persistent regimes of the atmospheric circulation. Given that weather system characteristics (such as positions, propagation directions, and displacements) are found to alter significantly with the breaking type, and that different North Atlantic weather regimes are largely characterized by different types of wave

breaking events, occurring in distinct geographical positions (e.g., Swenson and Straus, 2017), we expect the weather regimes to influence, and be influenced by, the cyclone and anticyclone life-cycles. The three-way interaction between the storm-tracks, RWBs, and the low-frequency flow representing the weather regime is left for further study, but initial results show that distinct and clearly preferred weather system paths, associated RWB positions, and resulting interactions with the low-frequency flow can be found for different weather regimes. An improved understanding of the relation between weather systems, RWB events,

and weather regimes can also help us improve our understanding of and confidence in projected future circulation changes (e.g., by relating changes in the frequency and positions of RWB events, storm-tracks, and the North-Atlantic jet).

*Data availability.* The ERA-Interim data used in this study has been obtained from the ECMWF data server: http://apps.ecmwf.int/datasets/

*Author contributions.* T.T.B. and N.H. designed the study and wrote the paper; T.T.B. performed the data analyses

*Competing interests.* At least one of the (co-)authors is a member of the editorial board of Weather and Climate Dynamics

*Acknowledgements.* The authors thank Chaim Garfinkel and Gudrun Magnusdottir for their help in providing the RWB detection algorithms, and to Kevin Hodges for providing the tracking algorithm. In addition, the authors wish to thank three anonymous reviewers, whose insightful comments and suggestions helped to considerably improve the manuscript. This research has been supported by the Israeli Science Foundation (ISF) Research Grants no. 1685/17 and 2713/17 of Prof. Nili Harnik.

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

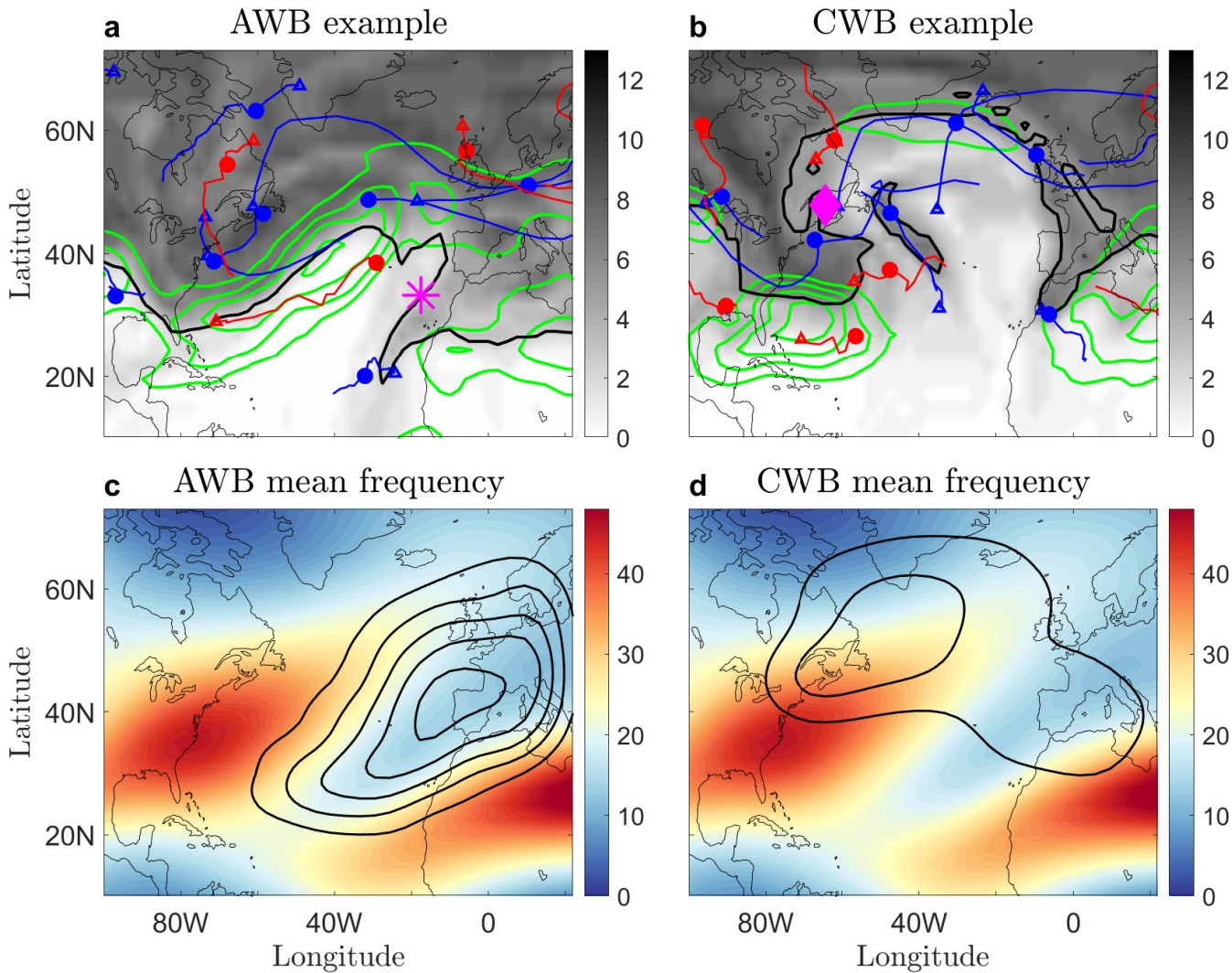

**Figure 1.** Examples of an (a) Anticyclonic Rossby Wave Breaking (AWB), and (b) Cyclonic Rossby Wave Breaking (CWB) event. Shown are the upper-level (250 hPa) Potential Vorticity (PV) in PV Units (PVU, 1 PVU=$10^{-6}$Kkg$^{-1}$m$^2$s$^{-1}$) (grey shading), the 250 hPa zonal flow U (green contours), and the low-level (850 hPa) tracks of cyclones (blue lines) and anticyclones (red lines), based on ERAI reanalysis data, for (a) Dec 7 1981 00UTC, and (b) Dec 11 1981 18UTC. The blue (red) dots denote the location of the cyclones (anticyclones) at the moment of the breaking, while the triangles denote the origin of the track. The centroid of the AWB (CWB) is denoted by a star (diamond), and the black line denotes the 2.5 (5.5) PVU contour. The lowest contour of U is equal to 30 m s$^{-1}$ and the contour spacing is 10 m s$^{-1}$. Panels (c) and (d) show the time-mean winter (DJF) upper-level (250 hPa) zonal flow U (colors) together with the Probability Density Functions (PDFs) (calculated using a kernel density estimator and multiplied by the number of events) of AWB and CWB centers, respectively, with lowest contour equal to 0.4 and contour intervals of 0.25. The PDF values denote the number of events per bin area (which is 4° in longitude and 1° in latitude in this case).

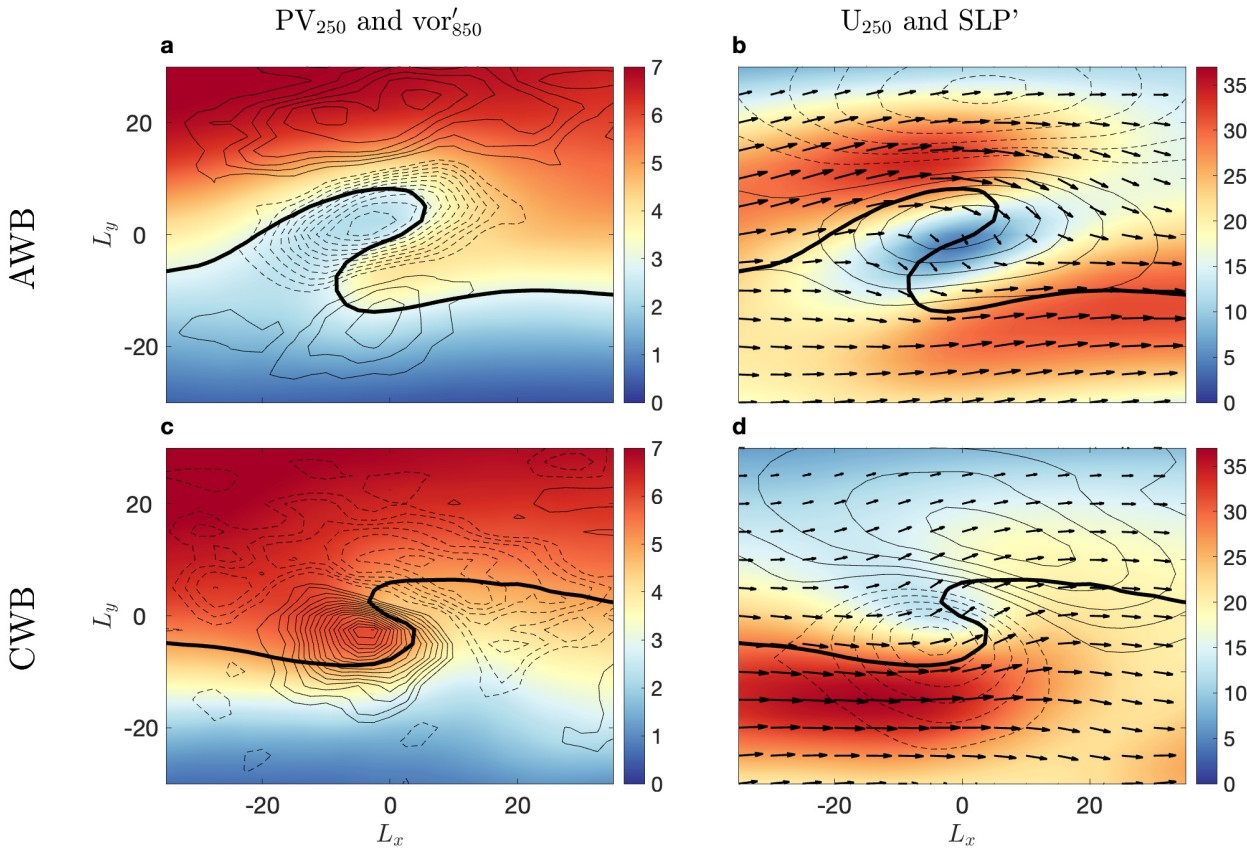

**Figure 2.** Composites of AWB (a,b) and CWB (c,d) events in the North Atlantic region, based on ERA-Interim reanalysis data, which occurred over the years 1980-2014 during December-February (DJF). Panels (a),(c) show the upper-level (250 hPa) PV field (in PVU, colors) and the 850 hPa vorticity anomaly (in $10^{-5}$ s$^{-1}$, black contours), while panels (b),(d) show the upper-level (250 hPa) zonal flow (in m s$^{-1}$, colors) and the SLP anomaly (in hPa, black contours), where the arrows denote the corresponding upper-level velocities. $L_y$ and $L_x$ denote the relative latitudinal and longitudinal distance (in degrees), respectively, from the center of the breaking. The black thick line in AWB (CWB) denotes the 3 (5) PVU contour. The lowest vorticity (SLP) anomaly contour is 0.15 (1), and the contour intervals are 0.1 (1).

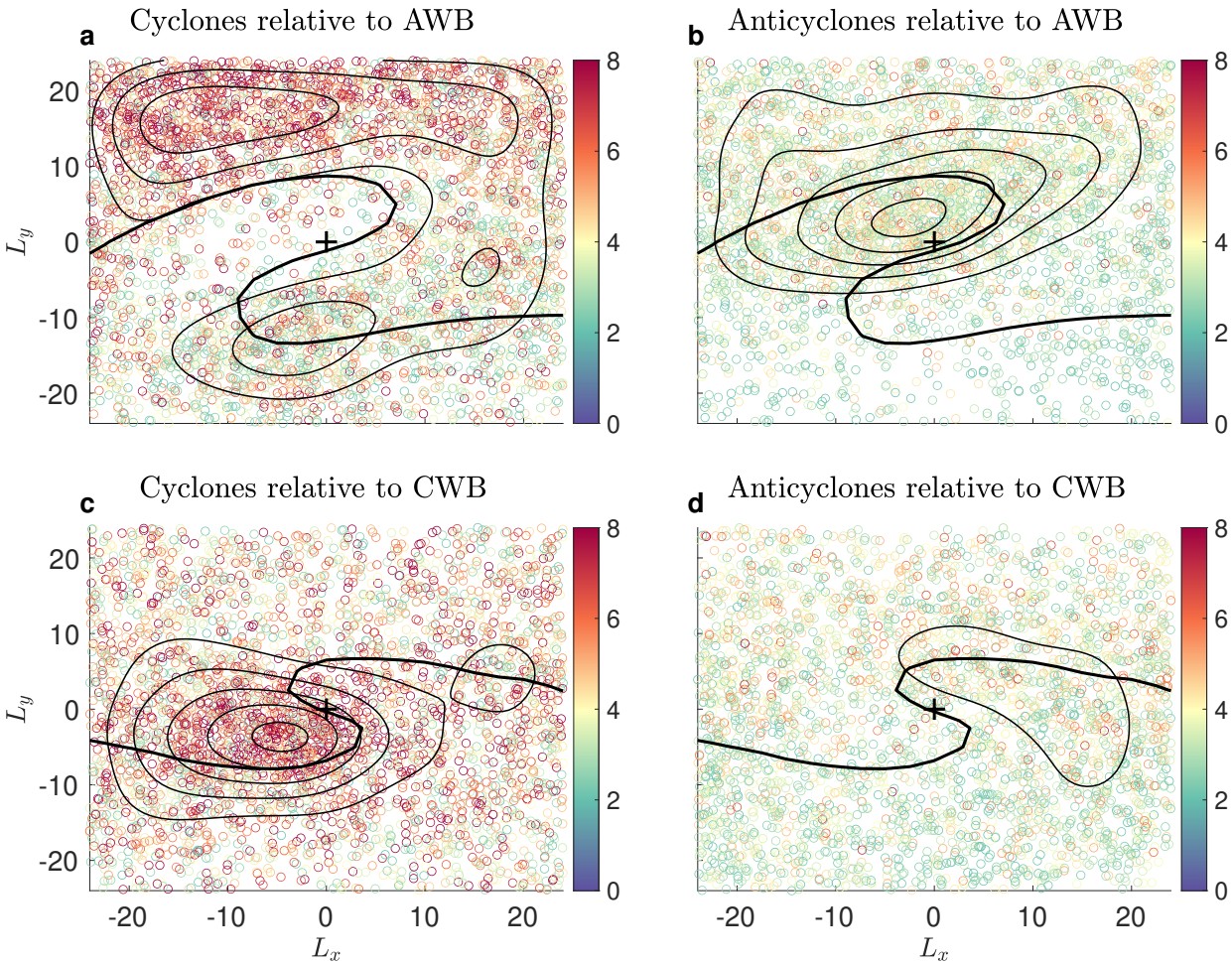

**Figure 3.** The positions of cyclones (a,c) and anticyclones (b,d) during AWB (first row) and CWB (second row), relative to the center of the RWB event (given by the cross symbol). Color indicates the intensity of the system (in absolute value) as identified by the tracking algorithm, in units of $10^{-5}\text{s}^{-1}$, and only systems with intensities larger than $2 \cdot 10^{-5}\text{s}^{-1}$ are plotted. $L_y$ and $L_x$ denote the relative latitudinal and longitudinal distance (in degrees), respectively, from the center of the breaking. The black thick line in the composites of AWB (CWB) denotes the 3 (5) PVU contour, while the thin lines denote the corresponding PDF (calculated using a kernel density estimator and multiplied by the number of systems) of the weather system centers, with lowest contour equal to 1.7 and contour intervals of 0.5. The PDF values denote the number of events per bin area (which is $0.6°$ in longitude and $0.6°$ in latitude in this case).

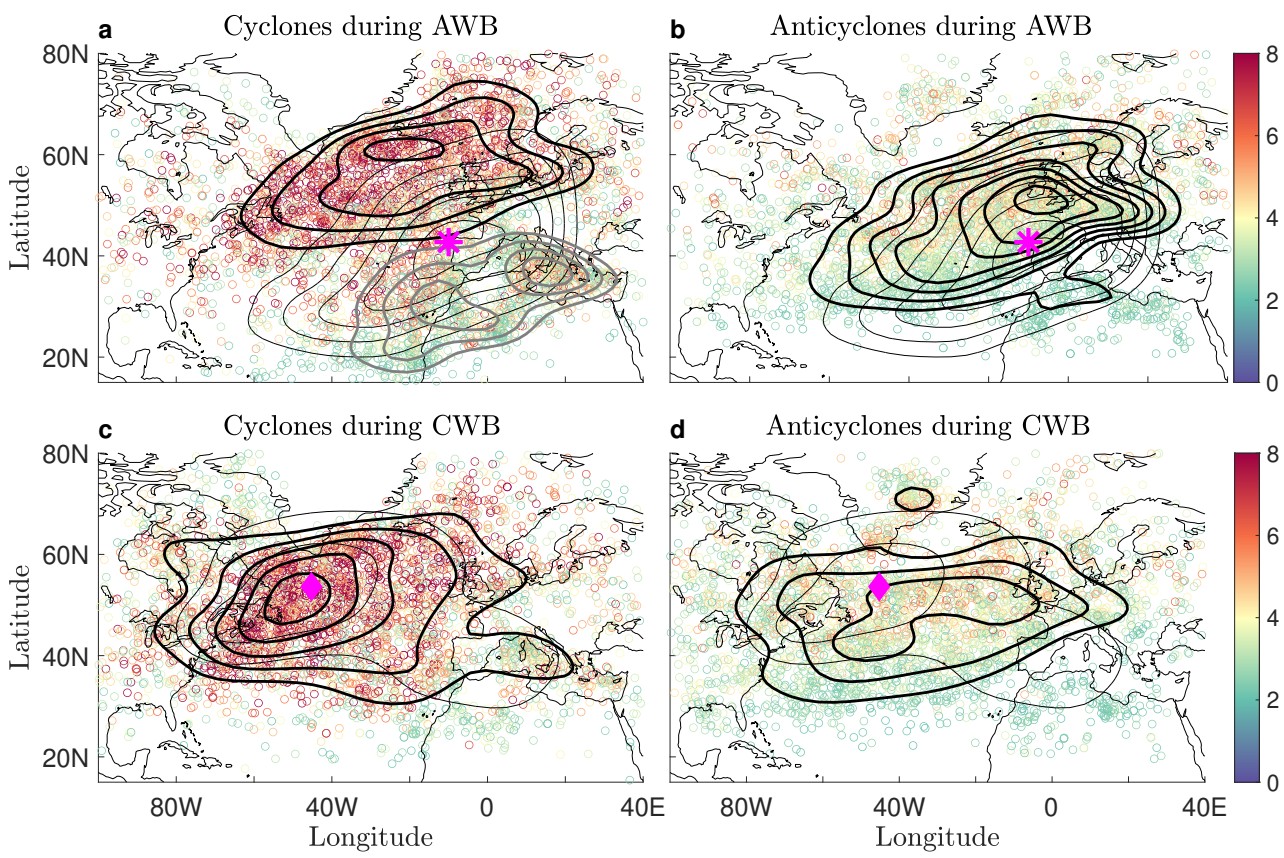

**Figure 4.** The actual spatial distributions of cyclones (a,c) and anticyclones (b,d) during AWB (first row) and CWB (second row), where color indicates the intensity of the system (in absolute value) in units of $10^{-5} \mathrm{s}^{-1}$, and only systems with intensities larger than $2 \cdot 10^{-5} \mathrm{s}^{-1}$ are plotted. Thick black contours denote the PDFs of the weather system counts (calculated using a kernel density estimator and multiplied by the number of systems), showing the locations where the weather systems are mostly found. In panel (a), the PDFs are separated into cyclones residing to the north of the AWB center (black thick contours) and cyclones residing to the south of the AWB center (gray thick contours). The lowest contour is equal to 0.6 and the contour intervals are 0.25. For reference, the thin black contours show the PDFs of AWB (panels a,b) and CWB (panels c,d) as in Fig. 1c,d, with lowest contour is equal to 0.4 and contour intervals equal to 0.25. The magenta star (diamond) denotes the location where AWB (CWB) centroids are most frequently found. The PDF values denote the number of events per bin area (which is $4°$ in longitude and $1°$ in latitude in this case). Overall, there are 5,106 cyclones during AWB, of which 2,922 to the north and 2,184 are to the south of the AWB center (panel a). In addition, there are 4,216 cyclones during CWBs (panel c), 4,131 anticyclones during AWB (panel b), and 3,681 anticyclones during CWB (panel d). The same systems and RWB events are used to produce Fig. 3.

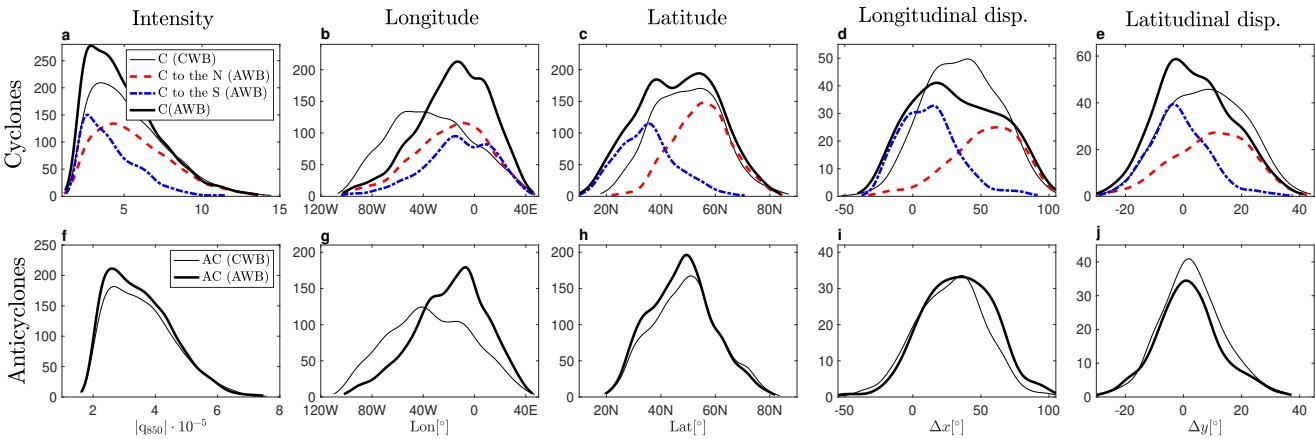

**Figure 5.** Surface weather system characteristics during AWB and CWB events. Fitted histogram lines showing the distribution of the intensity (absolute value, in units of $10^{-5}\text{s}^{-1}$) (a,f), longitudinal positions (in degrees) (b,g), latitudinal positions (in degrees) (c,h), longitudinal displacements (in degrees) (d,i) and latitudinal displacements (in degrees) (e,j) of the cyclones (first row) and anticyclones (second row) during RWB events, for the same systems shown in Fig. 3 and Fig. 4. For cyclones, these are separated into cyclones during CWB (thin black lines) and cyclones during AWB (thick black lines), where the latter are further separated into cyclones to the north (N) of AWB (red dashed lines) and cyclones to the south (S) of AWB (dotted dashed blue lines). For anticyclones, the separation is only between anticyclones during CWB (thin black lines) and anticyclones during AWB (thick black lines). The displacements are calculated as the difference between the position two days after the breaking, minus the position two days prior to the breaking, so only cyclones and anticyclones lasting for more than 5 days are used in this case (but similar results are found for displacements calculated in different manners). Hence, in panels (d),(i),(e) and (j), only 1,101 cyclones during AWB are used, where 635 cyclones reside to the north and 657 cyclones reside to the south of the AWB centroid. In addition, only 1292 cyclones during CWB, 640 anticyclones during AWB and 680 anticyclones during CWB are used.

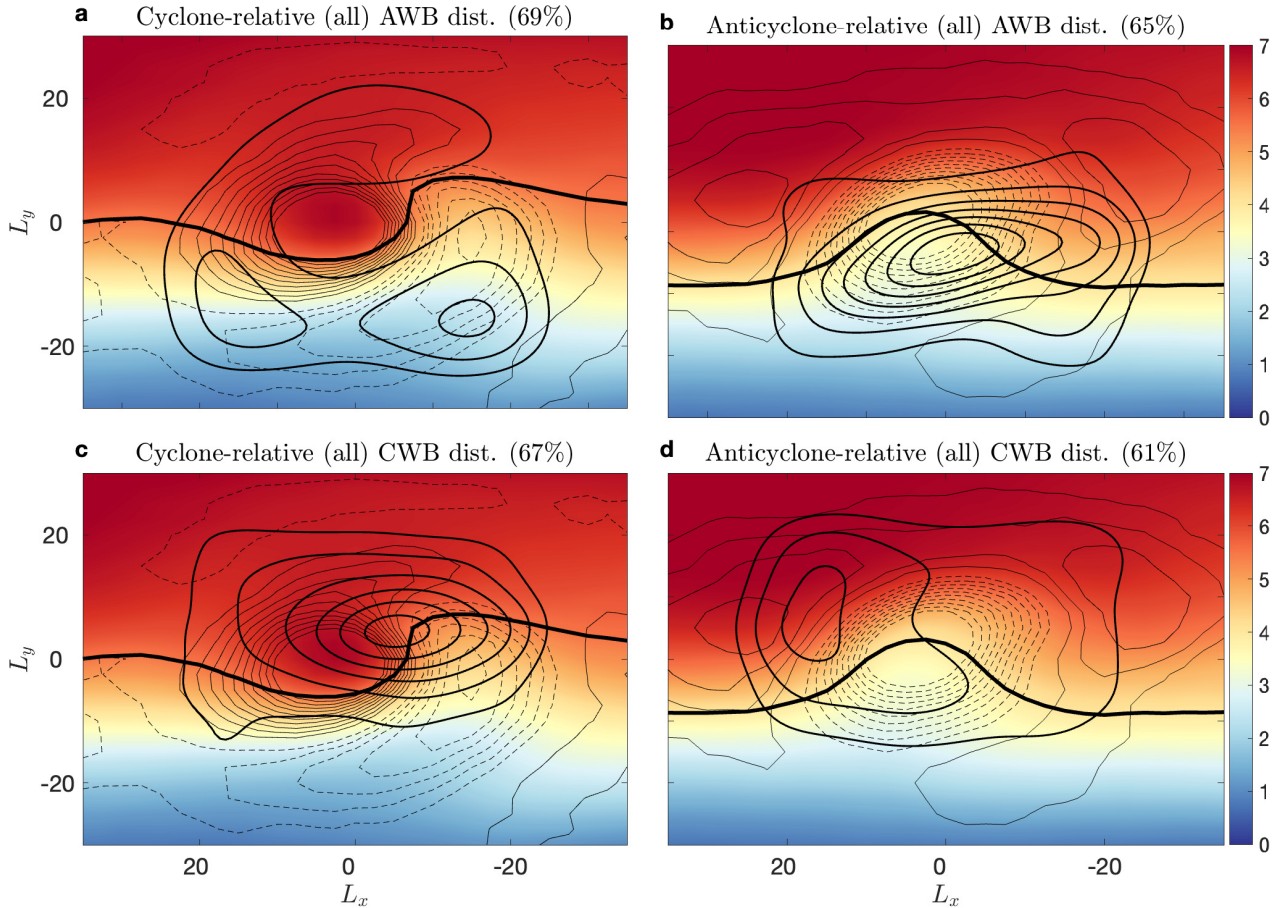

**Figure 6.** Weather-system relative composites, around all cyclones or all anticyclones in the North-Atlantic, overlaid with the relative RWB distributions. Upper-level (250 hPa) total PV (colors, in PVU) and PV anomaly (thin contours) centered on all cyclones (a,c) and anticyclones (b,d) in the North-Atlantic region (30N-60N and 80W-20E) at the time of maximum intensity. The black contours show the PDF of AWB (a,b) and CWB (c,d) relative to the center of the surface weather systems, with lowest contour equal to 0.5 and the contour intervals are 0.2. The PDF values denote the number of events per bin area (which is $0.6°$ in longitude and $0.6°$ in latitude in this case). $L_y$ and $L_x$ denote the relative latitudinal and longitudinal distance (in degrees), respectively, from the center of the weather systems. The thick black line denotes the 4.5 PVU contour in panels (a) and (c), while it denotes the 5.5 PVU contour in panels (b) and (d). The percentages in each panel denote the percentage of systems in each composite with the corresponding RWB sense occurring sometime during their lifetime.

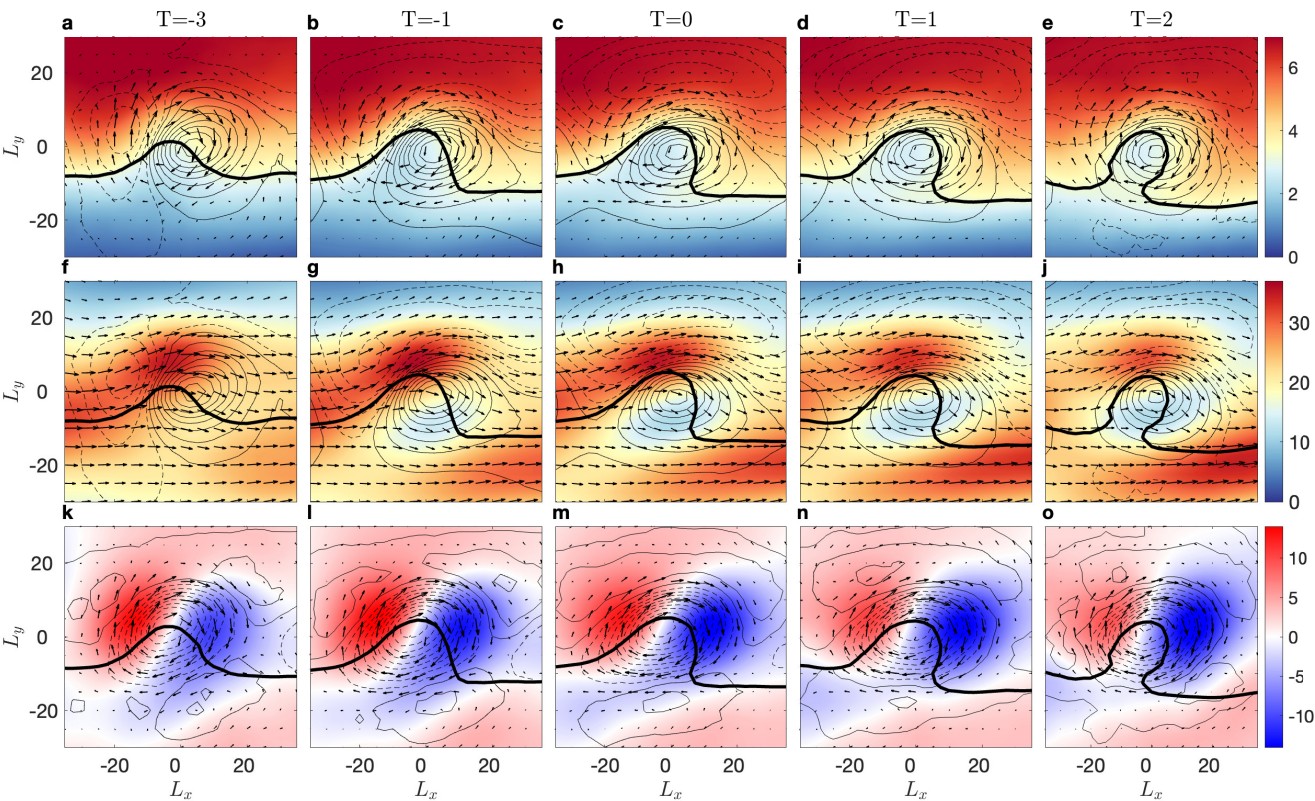

**Figure 7.** The same-sense anticyclone life cycle: Time lagged anticyclone-centered composites, for anticyclones occurring during AWB in the North Atlantic, from three days prior to the breaking ($T = -3$), and up to two days after the breaking ($T = 2$). Shown are the upper-level (250 hPa) PV in PVU (colors) and the SLP anomaly in hPa (contours) (first row), the upper-level (250 hPa) zonal wind in m s$^{-1}$ (colors) and SLP anomaly in hPa (contours) (second row), and the upper-level (250 hPa) meridional wind in m s$^{-1}$ (colors) and upper-level PV anomaly in PVU (contours) (third row). The arrows in panels (f-j) show the full upper-level velocities, while in panels (a-e) and (k-o) the anomalous upper-level velocities are shown. $L_y$ and $L_x$ denote the relative latitudinal and longitudinal distance (in degrees), respectively, from the center of the anticyclone. In all panels, the black line denotes the 3.2 PVU contour. The lowest contour for the SLP (PV) anomalies is 1 (0.1), while the contour interval is equal to 1 (0.2), where solid contours denote positive values and dashed contours denote negative values.

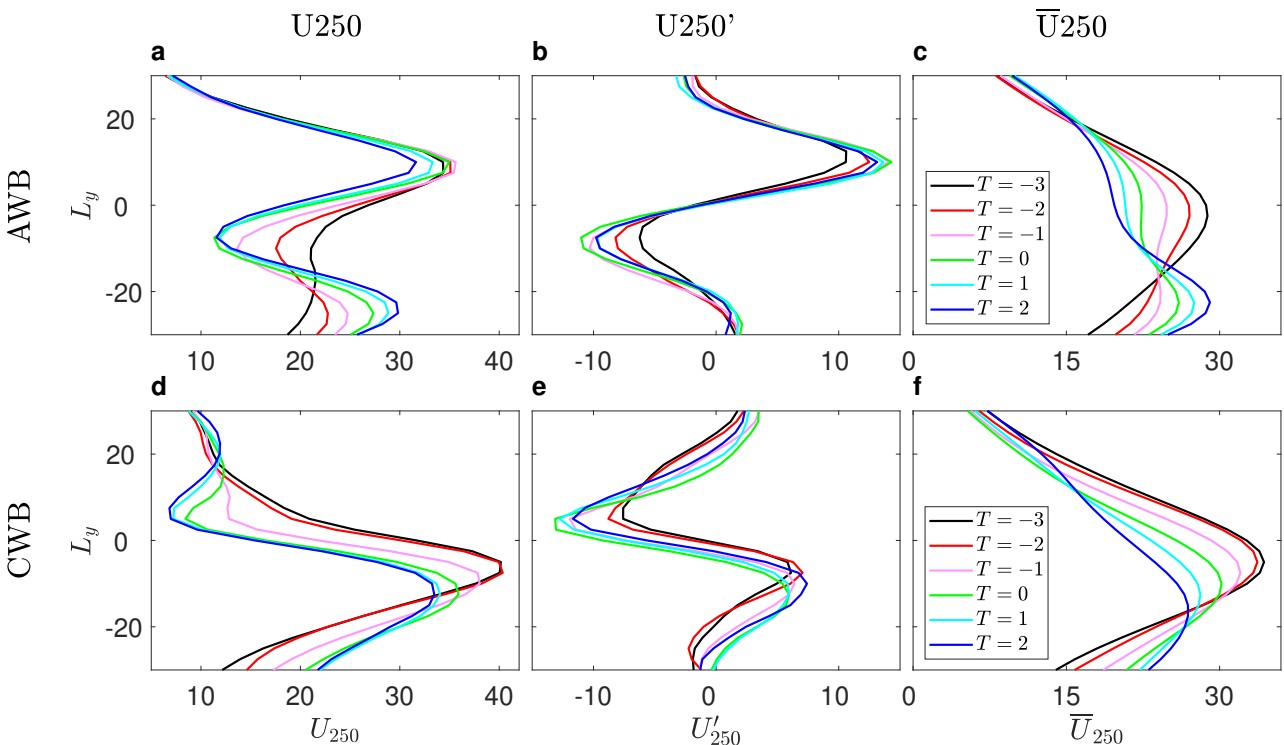

**Figure 8.** Cross sections of the composite upper-level (250 hPa) (a) total, (b) anomalous (deviation from time mean), and (c) climatological zonal wind (in units of m s$^{-1}$), at the longitude crossing the center of the anticyclones during AWB events. Panels (d), (e) and (f) show the same fields, but for cyclones during CWB events. The colors indicate the time, going from black (three days prior to the breaking), to blue (two days after the breaking). $L_y$ is the relative latitudinal distance (in degrees) from the center of the breaking.

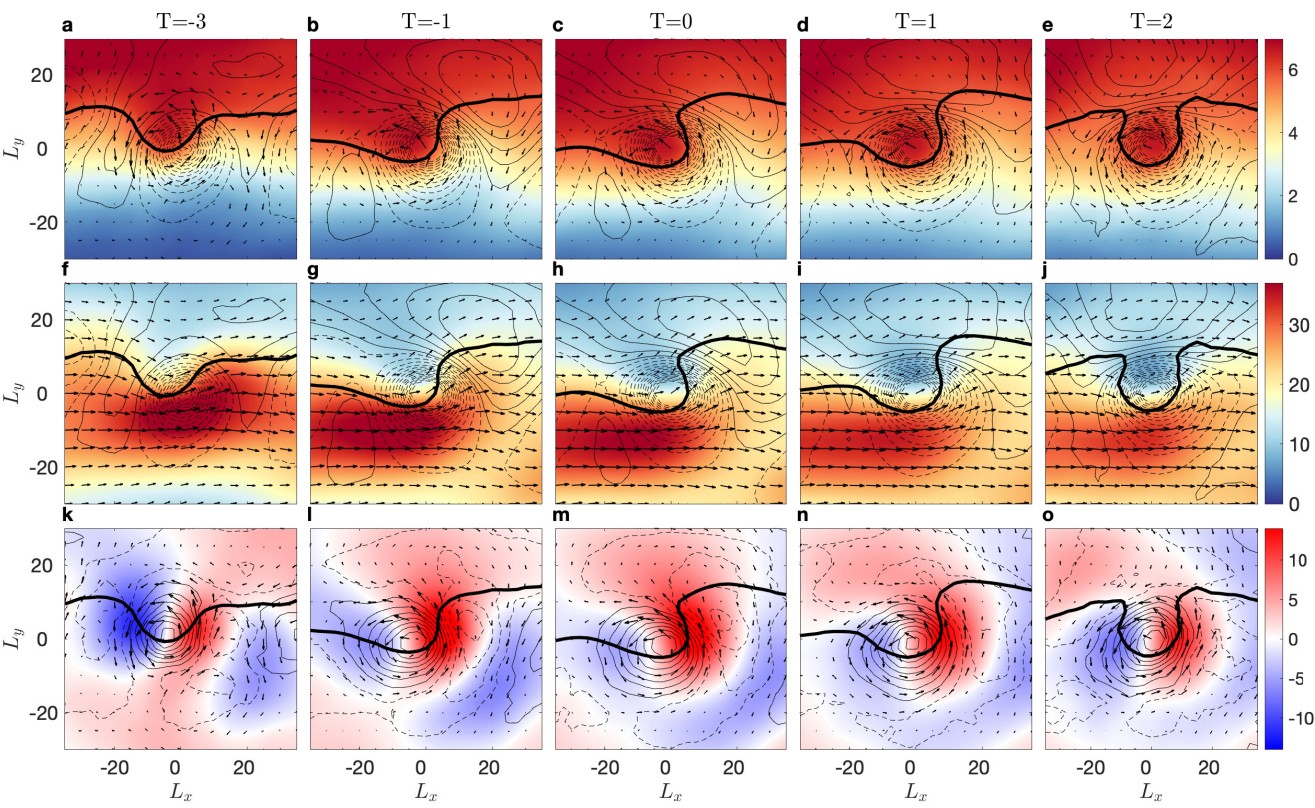

**Figure 9.** The same-sense cyclone life cycle: same as Fig. 7, but for composites centered on cyclones during CWB events, showing the time evolution of CWB in the North Atlantic region. In all panels, the black line denotes the 5.8 PVU contour.

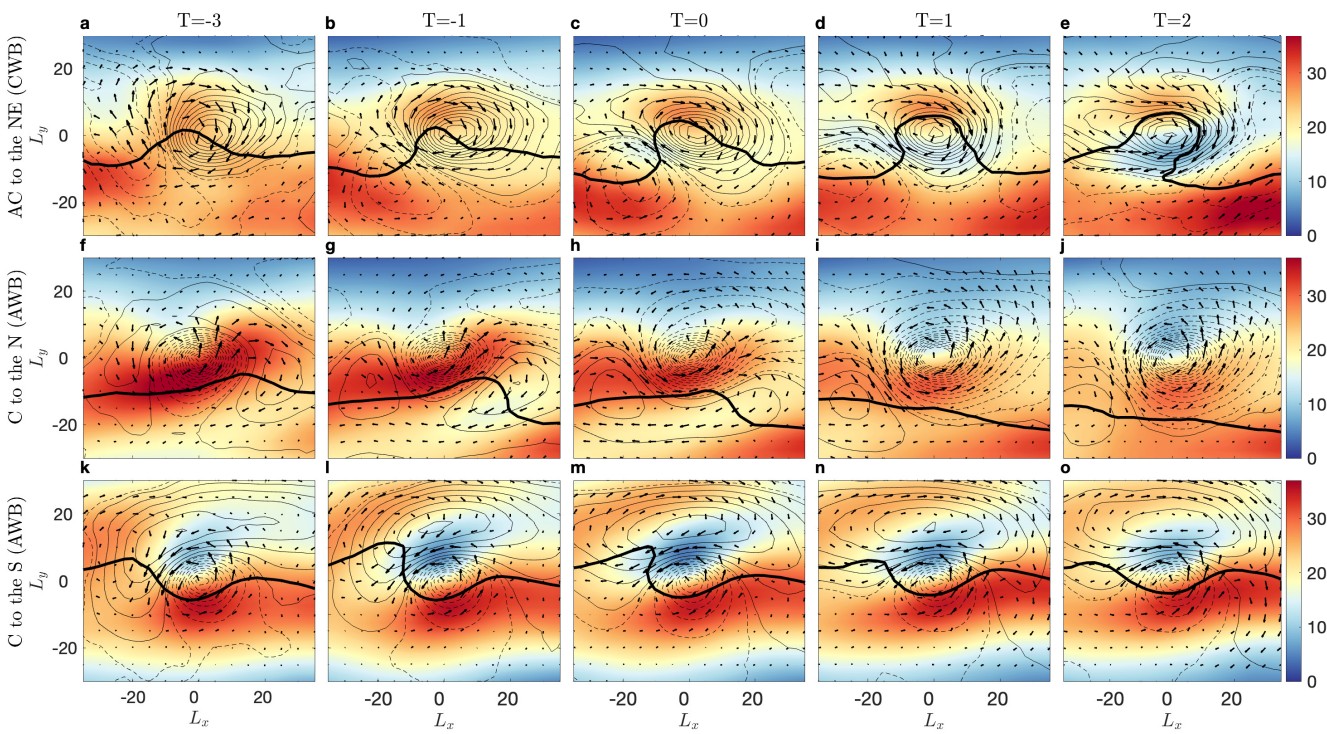

**Figure 10.** Opposite-sensed anticyclone and cyclone life cycles: composites centered on anticyclones residing to the northeast (NE) of the breaking center during CWB events (a-e), and around cyclones residing to the north (N) (f-j) and to the south (S) (k-o) of AWB in the North Atlantic region, from three days prior to the breaking ($T = -3$), and up to two day after the breaking ($T = 2$). Shown are the upper-level (250 hPa) zonal wind (colors) in m s$^{-1}$ and the SLP anomaly in hPa (contours), while the arrows denote the anomalous upper-level velocities. $L_y$ and $L_x$ denote the relative latitudinal and longitudinal distance (in degrees), respectively, from the center of the breaking. The black thick line denotes the 4.5 PVU contour in panels (a-e), the 3 PVU contour in panels (f-j), and the 3.5 PVU contour in panels (k-o).

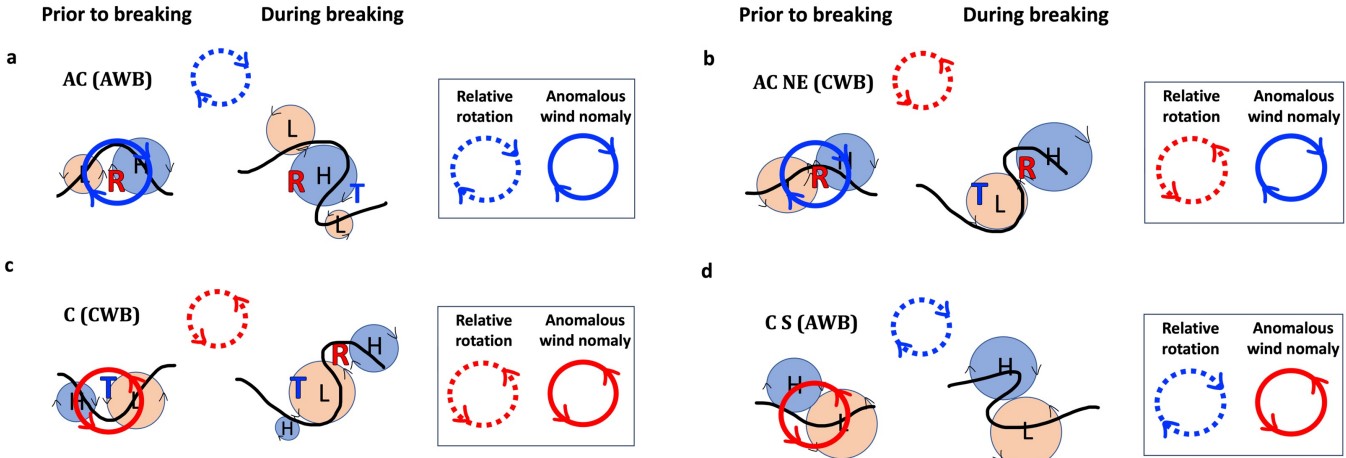

**Figure 11.** A schematic illustration showing the composite mean evolution of the upper- and lower-level circulation associated with anticyclones (AC, first row) and cyclones (C, second row) during AWB and CWB events. The left column corresponds to the "same-sense" weather system vorticity and RWB composites of (a) anticyclones during AWB events, and (c) cyclones during CWB events, while the right column corresponds to the "opposite-sense" weather system vorticity and RWB composites of (b) anticyclones to the North East (NE) of CWB events, and (d) cyclones to the South (S) of AWB events. The black solid line denotes a representative upper level PV contour in each case. The 'H' and 'L' symbols represent the surface anticyclone and cyclone, respectively, while the 'R' and 'T' symbols represent the upper-level ridge and trough, respectively. The solid red and blue circles represent the cyclonic and anticyclonic anomalous upper level winds, respectively, while the dashed red and blue circles represent the relative rotation between the pressure anomalies. The schematic summarizes the main composite life-cycle evolutions: for same-sense cases (a and c), the breaking and relative rotation are in the same sense as the weather system circulation, while for the opposite-sense cases (b and d), the breaking and relative rotation are in the opposite sense to the weather system circulation.