# Peer review of "The relation between Rossby Wave Breaking events and low-level weather systems"

_EGUsphere, 2023_

## Referee Comment (RC2)

This study examines the relation of surface cyclones and anticyclones and upper-tropospheric Rossby wave breaking. To do so, identification and tracking algorithms are applied on these systems and their relation examined using composites. A main result is the description of the (local) arrangement of anomalies in the vicinity of wave breaking that constitute latitudinal jet shifts in a larger-scale (e.g., zonal average) sense.

Overall, the manuscript is well prepared and straight forward to read. I have only minor comments regarding the presented analysis. My main reservation with this manuscript is that it did not become sufficiently clear to me how this work relates to previous work and the motivation of the study and the novelty of the insight remained somewhat unclear to me. After clarifying revisions, this manuscript will be well suited for publication in WCD.

**Main comment:**

**Relation to previous work: Motivation and novelty of insight**

A large body of literature exists on Rossby wave breaking (RWB) as part of a baroclinic life cycle (as noted by the authors in the introduction) and on the role of RWB in modifying the larger-scale jet pattern. I understand that bringing together tracked cyclones and identified RWB events has not been done in the way that it is done in this study. In this sense, there is an obvious novelty to the study. But scientifically, what are the open questions that are being addressed? In L114, the authors write: "Apart from the above mentioned studies, the intrinsic relation between the low-level cyclones, anticyclones, and RWB events has not been studied much, to the best of our knowledge. Here we highlight the fundamental relation between low-level weather systems and upper-level wave breaking events, focusing on the North Atlantic region." What is meant with "intrinsic" in the first sentence and with "fundamental" in the second? The first sentence disregards the large body of (synoptic-scale dynamics) literature that has studied baroclinic (cyclone) life cycles, which inherently involves the evolution of the upper-level trough, and work that has studied AWB (PV streamers) as precursors to cyclones. These studies often put case-study results in the context of conceptual/ idealized models. In what sense does the average over many real-atmospheric cases (with the caveats inherent in automated identification and processing of a large number of cases) yields more "intrinsic" and "fundamental" results than idealized experiments, which attempt to retain the essence of the problem? In what sense more "intrinsic" and "fundamental" results than an aggregation of decades of case studies? For the sake of the argument, I have phrased these questions somewhat provocatively. I do not mean to say that this study would not contain novel insight or make a valuable contribution. What I mean to say is that being (much) more specific about the open questions that motivate this study and about the new insights that this study contributes would be very helpful to better appreciate the authors' work.

A conclusion (or discussion) section usually gives a good opportunity to put results into context. The authors conclusions do not include a single reference to previous work. A quick look at Thorncroft et al. (1993) revealed that much of the described rearrangement of anomalies during RWB is consistent with that idealized study. Furthermore, the "anomalous" cyclones forming during AWB have extensively been studied previously also (more generally than in the context of Mediterranean cyclones only), and the authors results seem to be very much consistent with these studies. The reader needs more guidance to be able to identify the new insight generated by the current study, and this guidance should be given when summarizing the results, and not only when presenting specific results in the main body of the manuscript.

The authors highlight the modification of the jet pattern by rearrangement of the associated anomalies in a summarizing schematic. It should be noted that the arrangement of anomalies is a standard argument in the "blocking" community, i.e., to describe the weakening of the jet in the core of the block and the poleward/ equatorward deflection of the jet. RWB, jet structure and blocks are, of course, tightly related and the authors' composite most likely contain blocking situations also. The authors description put forth in the schematic thus seems to be a variant of a well-established argument.

**Minor comments:**

**Anomalous life cycle**
The authors introduce their definition of "anomalous" rather late in their manuscript and rather in passing. This leaves much room for confusion beforehand. Note that Simmons and Hoskins had denoted the cyclonic life cycle as "anomalous". In addition, the last paragraph in section 2 indicates that the "anomalous" cases are as frequent as other cases. I would thus suggest defining your meaning of the term at first use, and avoid using the term without definition, e.g., in the abstract.
I believe that the definition of "anomalous" is important (and non-standard) enough to move material from the supplement to the main text to introduce the definition. A more careful introduction could then include a discussion of existing knowledge of this "class" of cases (see main comment above. Note that there is a recent review paper on Mediterranean cyclones)

- Flaounas, E., Davolio, S., Raveh-Rubin, S., Pantillon, F., Miglietta, M.M., Gaertner, M.A., Hatzaki, M., Homar, V., Khodayar, S., Korres, G. and Kotroni, V., 2022. Mediterranean cyclones: Current knowledge and open questions on dynamics, prediction, climatology and impacts. *Weather and Climate Dynamics*, *3*(1), pp.173-208.

**Intensity of systems in the composite analysis**
The authors note in passing that the feature on which a composite is centered will be more intense than the other features in the composite, a well-known artefact of composites. Still, the authors use the intensity of composite systems in their subsequent arguments and conclusions. This is my main methodical issue with this study. A more careful analysis and discussion of the intensity of systems is needed. This could be based on benchmark composites of cyclones and anticyclones, and again I'd suggest in this case moving material from the supplement to the main text.

**L59**: Why do you not consider all four types of RWB? Please clarify.

**L240**: I am not sure I understand. What do you mean with "just signatures of the large-scale flow"? A barotropic Rossby-wave teleconnection pattern? Can you be more explicit about your possible alternative explanation/ hypothesis, such that the result that the anomalies are averages of synoptic-scale systems becomes more significant?

**L255**: In the average sense, a tripole emerges, but you do not analyze if the tripole does indeed co-occur often with AWB. Please substantiate the evidence or weaken your statement.

**L259**: I do not understand this paragraph. What is the additional information? Why switching back to physical space? Why do you not consider the pdf in the composite? Why now including

stronger systems only? Please clarify and provide more motivation for this approach. (Furthermore, note that several of your figure references have not been resolved.)

**L276**: From our understanding of cyclones that are associated with PV streamers (AWB) one would not expect them to propagate eastwards. I therefore do not see the concept of "hindering" appropriate for these cases.

**Section 4**: Does the analysis in this section use the closest (anti)cyclone as described in Sect. 2? Please clarify?

**Location of upper-level anomalies poleward and equatorward of the jet.**
The authors often refer to the location of anomalies relative to the jet. For a single jet, I would expect that upper-level anomalies are generated due to (synoptic-scale) deflections of the jet. In this simple case one would expect ridges to be located equatorward and troughs poleward of the (instantaneous) jet. In L309, the authors emphasize this seeming "standard" case with "importantly". How would the relative locations be reversed? Does it require a double-jet structure? Can the authors be more explicit about the significance of the relative location of the anomalies?

**L366**: I would always expect negative meridional velocity (northerly winds on the NH) between a ridge and trough downstream. In what sense is this velocity favored in this flow configuration?

**Section 5**: I have difficulties to see what we have learned from the „anomalous" life cycles. The section is mostly "show and tell". In the last paragraph of section 5.1: How would the shear enforce the trough? Evidently, we'd need some equatorward displacement of the PV contour to form the trough. How is the cyclonic shear doing this? How do we see the momentum fluxes? What, quite generally, do we learn in section 5.2? The statement in the last sentence of this section has long been realized and Mediterranean cyclones are a very active field of research.
In fact, the number of „anomalous" cases is of the same order of magnitude than other events. RWB looks quite different for the. „anomalous" cases. Can you clarify what we have for RWB? That RWB occurs with two distinct "flavors"?

e.g., **L494**, rotation of upper- and lower-level anomalies: I believe that work along the line of the following reference is relevant here.
  - Rivière, G., Arbogast, P., Lapeyre, G. and Maynard, K., 2012. A potential vorticity perspective on the motion of a mid-latitude winter storm. Geophysical Research Letters, 39(12).

**Second last sentence in the conclusions**: There is a large body of literature that examines the relation of RWB - or more generally scale interactions - and weather regimes. What specific aspects would the authors like to "examine more deeply"? What are the open questions that their approach could address? In the current version, the statement is so general that it seems rather meaningless.

Editorial comments:

**Abstract**

I did not find the abstract to be informative, because I could not identify the main motivation (open question) and main results of the study (see also main comment above).

**"Storm"-relative**

The authors describe composites centered on cyclones and anticyclones as "storm-relative". This misnomer did create some confusion while reading. I suggest finding a more appropriate wording.

**Use of supplementary material**

For me, for a research article – such as this manuscript is – there is too much use of supplementary material. Above, I have suggested how the manuscript can be strengthened and clarified by including some material in the main text. Personally, as a reader, I am usually confused by supplementary material. On page 11, e.g., there is a whole paragraph spent on describing material in the supplement. If this is important, why not including it in the main text? Do I need to consult the figure in the supplement? Or not? As a reader, I personally would like the authors to make this decision for me. I am aware, however, that I may be a minority with this opinion.

**L71ff**: I found this sentence hard to read. It seems that what you say is that Orlanski's results are consistent with the above, but from a PV flux perspective. I suggest simplifying the presentation here for readability (of the otherwise excellently written intro).

**L92ff**: In general, I dislike the use of parentheses to condense the presentation. (I know, I know, it is often used …). While this may be convenient for the authors, but as a reader I find it often very cumbersome to read. The specific sentence here contains many parentheses because you define also a few acronyms. Please consider rewording for improved readability. I'd further appreciate if you'd minimize use of parentheses to condense sentences throughout the text.

**L118-120**: Sounds like a contradiction. Please clarify.

**L233**: "breaking maximum in a relatively mature and developed stage". Stage of what? Wouldn't one expect a maximum being related to a mature and developed stage?

**L281ff**: Are all of the statements in this paragraph supported by a figure? Please clarify.

**L330**: Is this shown in Fig. S6 or in the main text? I was confused, please clarify.

**L505**: Why merely?

---

## Author Comment (AC1)

**Response to referee comments**

We thank all three reviewers for their helpful comments and suggestions. We are now working on a revised version of the manuscript, addressing all the reviewers' comments and concerns.

Most importantly, two of the reviewers asked to better phrase the motivation of the study, and the insights, significance and implications of the results, also in comparison with previous studies.
We agree with the reviewers that these aspects of the manuscript can be improved, and we are paying special attention in our revision to these important comments.
We prefer to address these issues fully when we submit the final response along with the revised manuscript, after we have completed the revision, to make sure we give a comprehensive and satisfying response.

Related to the above, Reviewer #2 also seems to suggest that we may have missed a large body of literature on three topics:
1. PV streamers during AWB as precursors to cyclones, more generally than in the context of Mediterranean cyclones only.
2. Baroclinic cyclone life cycles
3. The modification of the jet pattern by rearrangement of flow anomalies in blocking events

For (1), while we have found some relevant studies in the context of Mediterranean, tropical and subtropical cyclones (e.g., C. Davis , 2010: Simulations of Subtropical Cyclones in a Baroclinic Channel Model", https://doi.org/10.1175/2010JAS3411.1), we are not sure if the reviewer referred to midlatitude cyclones more generally. Note that the recent review paper on Mediterranean cyclones that the reviewer suggested mostly addressed these "class" of cyclones in the context of Mediterranean cyclones. It could be very helpful to know which papers exactly the reviewer is referring to, also in regard to (2).
Similarly, for (3), we are searching the literature and found some relevant studies (e.g., Yamazaki, A. and H. Itoh, 2009: Selective absorption mechanism for the maintenance of blocking. Geophys. Res. Lett., 36, L05803, doi:10.1029/2008GL036770), but not exactly what the reviewer seemed to suggest. Hence, it would be very helpful to get more specific references to the papers the reviewer was referring to.

We will submit our full response and revised manuscript soon and thank the reviewers again for their helpful comments.

Sincerely,
Talia Tamarin-Brodsky and Nili Harnik

---

## Author Response (AR1)

**Reviewer #1**

It is a well-written and interesting paper that contributes to the knowledge on Rossby wave breaking and their link with cyclones and anticyclones. The methodology is appropriate. I only have relatively minor remarks and technicalities that need to be addressed before publication. They are listed below following the order of the manuscript.

We thank the reviewer for the careful reading and very helpful comments.

*Main body of the manuscript:*

Line 67: In order to relate to the expressions given later in the same paragraph, maybe add (u'v') after "eddy momentum fluxes".

Fixed

Line 71: Please state that q is here the relative vorticity (and not the potential or absolute vorticity).

Added

Lines 142-143: Cyclones correspond to positive values of relative vorticity and anticyclones to negative values. Therefore, maxima in relative vorticity correspond to cyclones and minima to anticyclones. However, the threshold given here (10-5 s-1) is positive. Is it an absolute value of relative vorticity? The same comment applies for Fig. 3. The intensity represented with the colors and threshold given in the caption are always positive even for the anticyclones (panels b and d). Also, write somewhere that the intensity of the cyclones and anticyclones is given by the relative vorticity at 850 hPa at the cyclone centre location (and not an average or maximum value in an area around the cyclone).

The intensity values and threshold value are indeed in absolute values. We now explicitly mention this and added this information to the relevant figures as well. In addition, we now state that the intensity is based on the relative vorticity anomaly at the centre of the storm.

Line 155: Can the authors justify the use of the 350 K isentrope for detecting the RWB? Previous studies have used several isentropes to make sure that they cover the tropopause at all mid-latitudes (see for example Fig. 2a in Martius et al. (2007) (https://doi.org/10.1175/JAS3977.1) for the location of the tropopause relative to the isentropes). One can expect more (less) frequent AWB (CWB) at 350 K than at a lower isentrope. As the actual frequencies of AWB and CWB are not given in Fig. 1, it is difficult to get an idea of the relative frequency of the two wave breaking types

Thank you for this comment. The actual PDFs (calculated using a kernel density estimator and multiplied by the number of RWB events in case) are now plotted in Fig.1. We have added the suggested reference as well as a discussion on our choice for the 350K level. We acknowledge that RWB distributions depend on the vertical isentropic level chosen, with generally more (less) frequent AWB (CWB) at higher isentropic levels. The 350K isentropic surface, which corresponds approximately to the upper troposphere/lower stratosphere, has a relatively strong RWB activity. The choice of the 350K PV was motivated by earlier studies who used the same RWB

detection algorithm (Zhang, Magnusdotir). These studies concluded that the 350K level provide a useful representation of both AWB and CWB events over all latitudes, because higher latitude RWB events can be detected by higher PV values at this level. As noted in Zhang et al., higher vertical levels are often contaminated by pure stratospheric breaking, while lower vertical levels have considerably fewer events. In addition, the focus here is not on the relative frequencies of AWB and CWB events, but rather on the synotic-scale evolution of the flow leading to these events, which was similar when other vertical levels were examined.

We have added a short discussion about the choice of the 350K level for the RWB detection algorithm.

Line 166: How do the authors define "maximum overturning"? Please precise.

The maximum overturning here means the spatially largest overturning. This was added to the text.

Lines 166-168: How do the authors choose these three threshold values? Are they taken from a previous study?

In Strong and Magnusdotior, the longitudinal width of an overturning RWB is set to between 10-15 degrees, and the area of the breaking larger than 15*10^-4 as a fraction of earth's surface area. In our case, smaller values were taken in order to capture more RWB events (and smaller values were chosen for CWB, since they generally have significantly less events). However, we have tested many different values, and this did not influence any of our results qualitatively.

Lines 180, 227: I believe that "recovers" is not the right word in this context. I suggest to change this word's occurrences with "in agreement with" or "in accordance with" previous studies or a particular study. In lines 180-181, I suggest to cite Strong and Magnusdottir (2008) for example and maybe other studies.

We have changed "recovered" to "in accordance with previous studies" and added Strong and Magnusdottir (2008) and Zhang (2017) references.

Line 252: "anticyclone locations are mostly below the upper-level ridge". The meaning is not clear. I suggest to replace "below" with within.

We have changed the wording as suggested.

Lines 263-268: Only the cyclones positions are discussed. The anticyclones positions need to be discussed in this paragraph as well.

We have slightly revised the paragraph (lines 303-306).

Line 268: "maximum" -> most frequent.

We have changed the wording as suggested.

Line 283: Are the displacement values significantly different? Could the authors provide a standard deviation?

Following your comment, we have added a statistical significance test. The eastward displacements are indeed significantly different (statistically significant using a t-test at the 5% level). This has been added to the text.

The new means (when taking all anticyclones and not just the stronger ones) are 27.6 and 33.2 degrees for CWB and AWB, respectively (new Fig.5i). The corresponding standard deviations are 25 and 26 degrees (with 640 and 680 samples, correspondingly). The standard error of the mean (ERR=sqrt(sig1^2/N1+ sig2^2/N2) ) is thus ~1.4 degrees. This further information was not added to the text due to length consideration.

Lines 284-285: I do not find the statement about the anticyclones being located at higher latitudes convincing. First, because a reference position is needed: higher than what, than anticyclones positions during AWB or cyclones positions during CWB? Second, because apart from the local maximum over Greenland, the rest seems to be at lower latitudes than the cyclones during CWB and anticyclones during AWB. Moreover, the reference of Fig. 4 is missing to understand the full sentence.

We meant compared to anticyclones during AWB. Following your comment, we have added additional analysis to examine the PDFs of the longitudinal and latitudinal positions of cyclones and anticyclones during RWB events. Accordingly, we have added more panels to Fig.5 (new panels b,c and g,h).

As can be seen in the new Fig. 5, there is a clear longitudinal separation between the positions of cyclones and anticyclones. During CWB, both cyclones and anticyclones are generally more upstream compared to cyclones and anticyclones during AWB (Fig.5 b,g). Latitudinally, cyclones to the N of AWB are concentrated at higher latitudes compared to cyclones during CWB, while cyclones to the S of AWB are mostly to their south. The latitudinal distribution of anticyclones during CWB and AWB is similar, but anticyclones during CWB are slightly more concentrated to the north (difference is statistically significant at the 5% level).

For the propagation displacements, the differences between cyclones to the N and S of CWB are large and clear, as originally discussed in the manuscript. For anticyclones, the eastward propagation is slightly larger for anticyclones during AWB compared to anticyclones during CWB (difference is statistically significant at the 5% level, as mentioned in the previous comment), while the difference in the meridional displacements is not statistically significant.

Lines 303-305: This sentence is not very understandable to me and needs to be adjusted. Does the "in the next section" refer to section 5? It is not clear.

We have changed the wording to "in Section 5".

Line 335: "The negative upper-level meridional velocity … is mainly due to the intensifying anomalous ridge-trough system". How can the authors know that the southward meridional wind is due to the amplifying ridge-trough and not the cause for the amplification of the ridge and trough?

We have changed the wording to "consistent with". We did not try to imply causality here, just to highlight that the negative meridional velocity is getting more intense.

Line 340: It is not clear to me where the "background anticyclonic shear" is visible. Could the author precise it? Isn't there also cyclonic shear on the southern side of the anticyclone?

We have changed the wording to "the background anticyclonic time-mean jet" and referred to new Fig. S4 in the SI. It is true that there is also a cyclonic time-mean jet to the south of the anticyclone, however during the development stage of the breaking (e.g., Fig. S4a-b) the anticyclone is mostly influenced by the upstream anticyclonic shear.

Line 418: "an upper-level breaking in the corresponding sense is often found at upper-levels (Fig. S4 and Fig. S5)". To me, it is not clear from these two figures that there are any wave breaking happening. Therefore, I suggest to add the wave breaking frequency (as a percentage of the number of events considered in the composite and not normalised, see my previous comment above).

Thank you for this comment. We meant that an upper-level *rotation* in the corresponding sense is often found at upper-levels. However, we have now changed the text and no longer refer to these figures in the revised version (which are not included in the SI anymore). Instead, following your comment, we have added a new figure to the manuscript, showing composites of cyclones and anticyclones in the North Atlantic during the time of maximum intensity, overlaid with the RWB frequencies (see figure below).

This was done by aggregating the positions of AWB and CWB centers relative to the center of cyclones and anticyclones in the North Atlantic and calculating the corresponding PDFs (estimated as Kernel Density Estimators and multiplied by the number events in each case). In addition, we also quantify the percentage of cyclones or anticyclones that are associated with RWB in their vicinity (defined as less than 25 degrees to their north/south or east/west) sometime during their time evolution. Our results imply that in the North Atlantic, 69% of the cyclones are associated with an AWB in their vicinity, 67% with a CWB, while 11% do not have a RWB occurring in their vicinity during their lifetime (i.e. 47% of cyclones involve both types of wave breaking). For anticyclones, 65% are associated with an AWB in their vicinity, 61% with a CWB, while 15% do not have a RWB occurring in their vicinity during their lifetime (i.e. 41% of anticyclones involve both types of RWB). These percentages do not change much when taking only the 50 strongest storms from each season (not shown).

The results imply that most cyclones and anticyclones are involved with breaking at some point during their lifetime, with the largest subset having both types occurring in their vicinity during their life cycle. Interestingly, cyclones are slightly more associated with AWB (and not CWB). This is consistent with previous studies showing the AWB are generally more frequent. The AWB occurs mostly to the south-east of the cyclones, but the AWB relative positions are rather spread around the cyclone such that overall, the AWB frequency PDFs around cyclones are low. On the other hand, for cyclones associated with CWB, the breaking occurs in a similar position (close to their center and slightly to the north-east), hence the CWB frequency is high, even though there are generally fewer CWB events compared to AWB events.

[Figure]

**a**  AWB relative to cyclones (69%)  **b**  AWB relative to anticyclones (65%)

**c**  CWB relative to cyclones (67%)  **d**  CWB relative to anticyclones (61%)

A similar but opposite picture is found for anticyclones: for those associated with an AWB the breaking typically occurs in a similar relative position (close to their center and slightly to the south), hence the AWB frequency PDFs are high, while the CWB positions are more spread around the anticyclones (largely to their east or south-east), hence the frequency PDFs of CWB are lower. These results are consistent with our earlier results of Fig.3 from the manuscript.

The figure shows that composites of cyclones and anticyclones in fact mix between cyclonically and anticyclonically breaking storms. This motivates our further decomposition of cyclones and anticyclones into those breaking cyclonically and anticyclonically, to examine their distinct time evolution and characteristics. Following the comment of Reviewer #2, we no longer use the terminology of "anomalous", which can be misleading (since Thorncroft et al. used anomalous to describe the LC2 life cycle, and since our "anomalous" cases are as frequent as the canonical ones). Instead, we now denote the classification groups as "same-sense" breaking and storm sign (e.g., anticyclones with an AWB, and cyclones with a CWB), and "opposite-sense" breaking and storm sign (e.g., anticyclones with a CWB, and cyclones with an AWB).

We have added the above figure and discussion into the manuscript.

Section 5: Could the authors also display T = 1 and  T = 2 days on Fig. 9? It would help to see even more clearly how different these "anomalous life-cycles" are from the regular life cycles displayed in Figs. S4 and S5, which display T = -2 to T = 2 days.

This has been added as suggested (current figure 10). Note however that we have removed the SI figures showing the composites on all cyclones and anticyclones regardless of a wave breaking occuring (old version figs. S4 and S5), thus now  the "opoosite-sense  life cycles" should be

compared to current Figs 7 and 9 (old Figs. 6 and 8), which show composites over the storms during wave breaking events.

Lines 482 and 485: "while anticyclones are found to the NE of the cyclones" and "generally to the NE of the cyclones" seem to have the same meaning and the second occurrence to be an unnecessary repetition of the first. I suggest to remove the second occurrence.

Removed as suggested.

*Typos and others:*

Title: North-Atlantic -> North Atlantic

Line 37: (CWB),(e.g. Fig.1b) -> (CWB) (e.g. Fig. 1b)

Line 60: Section 22.2 -> Section 2.2

Line 129: Conclusion -> Conclusions

Line 135: December-January -> December-January-February

Lines 132-133: velocities -> wind

Line 142: 850 hPa vorticity -> 850 hPa relative vorticity

Line 143: centeres -> centers (US spelling) (or centres UK). "centers" is used on line 147.

Line 143: 1.10-5 -> 10-5. The 1 does not seem useful.

Line 176: centeroid -> centroid

Line 178: braking -> breaking

Line 198: Full stop missing at the end of the sentence.

Line 199: storm-ceneterd -> storm-centered

Lines 216, 223: show -> shows

Line 223: a low-PV… a high-PV -> a low-PV tongue … a high-PV tongue

Line 225: deccelerate -> decelerate

Line 226: composite -> composites

Lines 260, 262, 263, 264, 267: "??" The references to Fig. 4 did not appear well.

Lines 261-262: magnitudes -> intensities

Line 9 and throughout the manuscript: I do not think it is correct to "centre around". I would rather say "centre on" the (anti)cyclone location.

Line 303: subsection -> subsections

Lines 343-344: Tamarin and Kaspi, 2016 -> Tamarin and Kaspi (2016)

Line 368: anticycloniclly -> anticyclonically

Line 402: Fixing -> Adjusting

Line 406: jet is remains -> jet remains

Line 411: it east -> its east

Line 415: In section (4a) and (4b) -> In sections 4.1 and 4.2

Line 460: Fig. ??a. The reference to the figure (Fig. 4?) did not work.

Line 520: seem -> seems

References: Almost all capital letters are missing in the titles of the referenced articles, the journal names should be abbreviated and the doi provided (see https://www.weather-climate-dynamics.net/submission.html#references). For example, in the first reference of Benedict et al., north atlantic should be North Atlantic, Journal of the Atmospheric Sciences should be abbreviated, and the doi provided. The issue number is not necessary (for example in Eady 1949).

Figure 4: In the sentence before the last: denote -> denotes

Figure S8: "after to" -> after

Overall, I suggest to replace "velocity" with air velocity, wind, or wind speed.

All the above typos including the references have been corrected as suggested.

*Figures:*

Several figures display PV at 250 hPa whereas, RWB events are detected in PV fields on the 350-K isentrope (line 155). Although the authors wrote that the results are similar when detecting RWB in PV at 250 hPa and use the wind at 250 hPa, I still find it inconsistent to display PV at 250 hPa and not at 350 K since it is the field used to detect RWB.

We completely understand the reviewer's concern. However, note that it is very common to detect RWB on isentropic levels and then plot the atmospheric variables on other surfaces such as SLP or low-level winds (e.g., Strong and Magnusdottir, 2008; Zhang and Wang, 2018). Since the focus here is also on the low-level weather systems, and we are tracking vorticity on the 850 hPa pressure level, this seems to us like the most suitable surface for analyzing the circulation, and therefore we also prefer to compare it with the 250 hPa level circulation. To address your

concern, however, we have added here a comparison of the PV at 250 hPa and at the 350-K isentropic level. As you can see, the two surfaces yield similar results. We have added a comment discussing this point.

[Figure]

About the cyclone-centred figures: it looks like a "random" PV isoline is plotted. I assume that the contour value has to be changed to illustrate the authors' point, but the reason for using different contours on different figures could be stated. Also, as an example why choosing the 3.2 PVU contour in Fig. 6? Was the 3 PVU contour not good enough?

We have indeed chosen different contours in each plot to highlight or illustrate certain aspects of the flow. For example, in Fig.6 (now Fig.7), the 3 PVU contour shows a closed PV contour in the anticyclone center (by T=2), which we worried might be confusing for the reader and therefore decided not to use it. The 3.5 PVU, on the other hand, did not show an overturning of the PV contour, which is why we ultimately chose the 3.2 PVU contour. We do not find it necessary to include this reasoning in the text, but did change the values used to be more consistent wherever possible (for example in Fig.3 which indeed should be consistent with Fig.2).

Figure 1: I understand why the authors are representing the wave breaking frequencies with normalised pdfs, but information about the relative (and actual) frequency of the two types is lost, isn't it? Or do AWB and CWB have the same frequency? The year of the two particular events (panels a and b) is not given in the caption. Please add it and maybe save space by shortening the dates such as, e.g., 7 Dec YEAR 00UTC. Finally, I suggest to increase the size of the domain to the east, such as to be more consistent with the domain mentioned in line 171, that extends to 20ºE (starting the figure at 260ºE is fine). It would also be great if the longitude values in the x-axis could correspond to the values in the text (expressed with ºW), that is instead of "270, 300, 330, 360", -90, -60 -30 and 0 could be used.

We have replaced the normalized PDFs in Fig.1 with the actual PDFs (calculated using a kernel density estimator and multiplied by the number of events in each case). The year of the two

particular events is now clearly stated and we have shortened the date as suggested. All the other suggestions have been incorporated as well.

Figure 2c: How do the authors explain that the low-PV streamer is not as distinct as for the AWB composite? A blue tongue is not present.

A blue tongue is not present because CWB events occur more poleward, and therefore on higher PV contour values. The horizontal difference of PV in the two types of breakings, for example, is similar (around 1.5-2 PVU). The color scheme happens to highlight better the poleward PV tongue in the AWB compared to CWB.

Figure 3: See my comment above about the intensity of anticyclones. Anticyclones should have negative relative vorticity but the colorbar shows positive values for both cyclones and anticyclones. Could the author mention that the absolute relative vorticity is used (if that is indeed the case)? Also, why is the PV contour not the same as in Fig. 2? Aren't they composites over the same wave breaking events?

The color indicates the relative vorticity intensity of the system in absolute value. This has been added to the caption. There was some mix-up with the values used and those indicated in the paper. Both figures now use the same PV contours (3 and 5 PVU) and the captions have been corrected.

Figure 5: It would be great to have x-axis labels. Moreover, the name of the second row needs to be changed: Antiyclones -> Anticyclones

Thank you, this has been corrected.

Figure 7: The letters designating the panels a, b, and d are not well aligned. I would remove the Ucross as x-axis label and replace them with the columns title. I would change "U250 tav" to U250 with a bar over U to be consistent with the notation in the text (line 329). If the authors make this change, they can also change "U250 total" to U250. The red and pink lines are not very well distinguishable. A lighter pink would be maybe better.

This has been corrected as suggested.

Figure 8: "Same as Fig. 4". Isn't this figure the same as Fig. 6?

Thank you, this has been corrected.

Figure 9: On the top label of the middle figure it should be T=-1 and not T=1.

Thank you, this has been corrected. Note that we are now showing the time evolution during: T=-3, T=-1, T=0, T=1, and T=2, so previous Fig. 9 (now Fig.10) is consistent with previous Figs.6 and 8.

Units in captions and text:

- the authors often use ms-1 as unit for wind. I personally find this writing misleading as it can refer to millisecond and not meters per second. Therefore, I suggest to add a white space in between the m and the s: m s$^{-1}$.

This has been corrected as suggested.

- the authors use both mb and hPa as unit for pressure. Choose one and keep it all along the manuscript.

 This has been corrected as suggested (we now use only hPa)

*Supplement:*

Figure S2: It is similar to Fig. 4 of the manuscript and not to Fig. 3.

We suspect the reviewer may be mistaken here- Fig.3 in the manuscript indeed shows the relative positions of storms, similar to Fig.S2.

Figure S3: The same colormap as Fig. 1c,d could be used for an easier comparison.

This has been corrected as suggested.

Figures S4, S5 a,f: Do these figures actually show T=-2 days and not T=-3 days? In the manuscript T=-3 days is used. Moreover, please state that the PV anomalies are the contours (top row), the zonal wind at 250 hPa the shading and the SLP anomaly the contours (bottom row).

In the current version of the manuscript, we have decided not to include Fig.S4 and Fig.S5 in the SI, instead we added new Fig.6 to the manuscript.

Figure S5: This figure is the same as Fig. S4 and not S1.

Thank you. However, in the current version of the manuscript we have decided not to include Fig.S4 and Fig.S5.

Figures S6 and S7: The title of panel a shows T=-3, but the caption says T=-2 days. Please correct.

This should have been T=-3, this has been corrected.

Figure S7: This figure is the same as Fig. S6 and not S3.

Thank you, this has been corrected.

**Reviewer #2**

This study examines the relation of surface cyclones and anticyclones and upper-tropospheric Rossby wave breaking. To do so, identification and tracking algorithms are applied on these systems and their relation examined using composites. A main result is the description of the (local) arrangement of anomalies in the vicinity of wave breaking that constitute latitudinal jet shifts in a larger-scale (e.g., zonal average) sense. Overall, the manuscript is well prepared and straight forward to read. I have only minor comments regarding the presented analysis. My main reservation with this manuscript is that it did not become sufficiently clear to me how this work relates to previous work and the motivation of the study and the novelty of the insight remained somewhat unclear to me. After clarifying revisions, this manuscript will be well suited for publication in WCD.

We thank the reviewer for the careful reading and very helpful comments.

Main comment:

Relation to previous work: Motivation and novelty of insight

A large body of literature exists on Rossby wave breaking (RWB) as part of a baroclinic life cycle (as noted by the authors in the introduction) and on the role of RWB in modifying the larger-scale jet pattern. I understand that bringing together tracked cyclones and identified RWB events has not been done in the way that it is done in this study. In this sense, there is an obvious novelty to the study. But scientifically, what are the open questions that are being addressed? In L114, the authors write: "Apart from the above mentioned studies, the intrinsic relation between the low-level cyclones, anticyclones, and RWB events has not been studied much, to the best of our knowledge. Here we highlight the fundamental relation between low level weather systems and upper-level wave breaking events, focusing on the North Atlantic region." What is meant with "intrinsic" in the first sentence and with "fundamental" in the second? The first sentence disregards the large body of (synoptic-scale dynamics) literature that has studied baroclinic (cyclone) life cycles, which inherently involves the evolution of the upper-level trough, and work that has studied AWB (PV streamers) as precursors to cyclones. These studies often put case-study results in the context of conceptual/ idealized models. In what sense does the average over many real-atmospheric cases (with the caveats inherent in automated identification and processing of a large number of cases) yields more "intrinsic" and "fundamental" results than idealized experiments, which attempt to retain the essence of the problem? In what sense more "intrinsic" and "fundamental" results than an aggregation of decades of case studies? For the sake of the argument, I have phrased these questions somewhat provocatively. I do not mean to say that this study would not contain novel insight or make a valuable contribution. What I mean to say is that being (much) more specific about the open questions that motivate this study and about the new insights that this study contributes would be very helpful to better appreciate the authors' work.

Thank you for these important comments. We have now modified the title, abstract, introduction, summary, and main body of the text (where relevant) to better convey our motivation for the current analysis, the open questions, the novelty and insights of our results and relation to previous studies. This included restructuring of old sections 4-5 into a single section 4 and changing the terminology to clarify our analysis and the novelty of our results.

By "intrinsic" and "fundamental" we did not mean that our composite analysis yields more basic results than idealized studies or decades of aggregated case studies. Rather, we meant that the upper-level wave breakings and the storms are inherently related (as the reviewer suggested as well). However, to avoid confusion, we have omitted the word "intrinsic" from the title and text, as well as "fundamental" and similar claims. We have also revised the text and added more references that discuss both case studies and AWB as precursors to cyclones. We would greatly appreciate any further references the reviewer might have in mind that we may have missed.

We agree with the reviewer that the relation between storms and RWB events has been known for decades, and we now highlight it more clearly in the text. The motivation for the current study is to highlight the following open questions:

- What is the relation between RWB events and low-level weather systems? For example, do RWBs and weather system always occur simultaneously? What are the percentages of storms involved with each type of breaking, and vice versa?
- How do storm characteristics (including geographical positions, intensity, and displacements) and the composite time evolution differ, depending on the type of upper-level RWB and their position relative to the breaking?
- How well do idealized life cycle experiments, which use a specified initial perturbation with a single zonal wavenumber and a prescribed simplified initial zonal jet, capture the life cycle of real-atmosphere cyclones and anticyclones?

While examining case studies can be very insightful, some of these questions cannot be addressed based on individual cases alone. Using an automated detection algorithm of RWBs and storms can therefore supplement these existing studies and generalize their results.

In a broader sense, a better understanding of the different lifecycles of real-atmosphere storms and the upper-level breaking they involve, can be useful, for example, for studying extremes, for improving weather predictability, for exploring the relation between storm tracks and slowly varying weather regimes and how it is mediated by RWB events, and for improving our confidence in projected future circulation changes (e.g., by relating changes in the frequency and positions of RWB events, storm tracks, and the North-Atlantic jet).

All of the above discussion has been added to the text to better convey our broader motivation, and the open questions specifically addressed.

A conclusion (or discussion) section usually gives a good opportunity to put results into context. The authors conclusions do not include a single reference to previous work. A quick look at Thorncroft et al. (1993) revealed that much of the described rearrangement of anomalies during RWB is consistent with that idealized study. Furthermore, the "anomalous" cyclones forming during AWB have extensively been studied previously also (more generally than in the context of Mediterranean cyclones only), and the authors results seem to be very much consistent with these studies. The reader needs more guidance to be able to identify the new insight generated by the current study, and this guidance should be given when summarizing the results, and not only when presenting specific results in the main body of the manuscript.

We have added references to the conclusions section and revised the text to put our results more into context and to highlight the new aspects of our results. The reviewer is referred to the new summary in the revised version.

The authors highlight the modification of the jet pattern by rearrangement of the associated anomalies in a summarizing schematic. It should be noted that the arrangement of anomalies is a standard argument in the "blocking" community, i.e., to describe the weakening of the jet in the core of the block and the poleward/ equatorward deflection of the jet. RWB, jet structure and blocks are, of course, tightly related and the authors' composite most likely contain blocking situations also. The authors description put forth in the schematic thus seems to be a variant of a well-established argument.

We have modified the schematic shown in new Fig.11 and now include also the "opposite-sense" cases. The motivation of the new Fig.11 is to emphasize different potential time-evolutions of cyclones and anticyclones, depending on the RWB they are associated with (and their position relative to the RWB center).

Minor comments:

Anomalous life cycle

The authors introduce their definition of "anomalous" rather late in their manuscript and rather in passing. This leaves much room for confusion beforehand. Note that Simmons and Hoskins had denoted the cyclonic life cycle as "anomalous". In addition, the last paragraph in section 2 indicates that the "anomalous" cases are as frequent as other cases. I would thus suggest defining your meaning of the term at first use, and avoid using the term without definition, e.g., in the abstract.

I believe that the definition of "anomalous" is important (and non-standard) enough to move material from the supplement to the main text to introduce the definition. A more careful introduction could then include a discussion of existing knowledge of this "class" of cases (see main comment above. Note that there is a recent review paper on Mediterranean cyclones)

- Flaounas, E., Davolio, S., Raveh-Rubin, S., Pantillon, F., Miglietta, M.M., Gaertner, M.A., Hatzaki, M., Homar, V., Khodayar, S., Korres, G. and Kotroni, V., 2022. Mediterranean cyclones: Current knowledge and open questions on dynamics, prediction, climatology and impacts. Weather and Climate Dynamics, 3(1), pp.173-208.

Thank you for this comment and reference. The suggested paper and other relevant papers have been added to the manuscript in the context of this class of cyclones (in the introduction and other places). Following your comment and Reviewer's #1 comment on previous Figs. S4 and S5, we also no longer use the terminology "anomalous", which can be misleading. Instead, we now use a terminology of "same-sense" storm vorticity and breaking type (e.g., anticyclones with an AWB, and cyclones with a CWB), and "opposite-sense" storm vorticity and breaking type (e.g., anticyclones with a CWB, and cyclones with an AWB).

In addition, Fig.S4 and Fig.S5 from the previous version are no longer included in neither the SI nor the manuscript. Instead, we have replaced them with new Fig.6 in the manuscript. The figure shows, as suggested by Reviewer #1, the composites of cyclones and anticyclones in the North Atlantic during their time of maximum intensity, overlaid with the RWB frequencies (see Figure 1 below). The RWB frequencies were calculated by aggregating the positions of AWB and CWB centers relative to the center of the storms and calculating the corresponding PDFs (estimated as Kernel Density Estimators and multiplied by the number events in each case). We

[Figure]

**a** AWB relative to cyclones (69%)   **b** AWB relative to anticyclones (65%)

**c** CWB relative to cyclones (67%)   **d** CWB relative to anticyclones (61%)

also quantify the percentage of cyclones or anticyclones that are associated with RWB in their vicinity (defined as less than 25 degrees to their north/south or east/west) sometime during their time evolution. Our results imply that in the North Atlantic, 69% of the cyclones are associated with an AWB in their vicinity, 67% with a CWB, while 11% do not have a RWB occurring in their vicinity during their lifetime. For anticyclones, 65% are associated with an AWB in their vicinity, 61% with a CWB, while 15% do not have a RWB occurring in their vicinity during their lifetime. These percentages do not change much when taking only the 50 strongest storms from each season (not shown).

The results imply that most cyclones and anticyclones are involved with breaking at some point during their lifetime, with the largest subset invovling both CWB and AWB in their vicinity during their life cycle. Interestingly, the largest pairing, percentagewise is between cyclones and AWB. The existence of more AWB compared to CWB associated with cyclones, makes sense given that there is overall more AWB compared to CWB, but is probably also related to the fact that diabatic heating associated with the warm conveyer belt contributes to upper-level ridge development. The AWB occurs mostly to the south-east of the cyclones, but the AWB relative positions are rather spread around the cyclone such that overall, the AWB frequency PDF around cyclones is low. On the other hand, for cyclones associated with CWB, the breaking occurs in a similar position (close to their center and slightly to the north-east), hence the CWB frequency PDF is high (even though there are generally fewer CWB events compared to AWB events).

A similar but opposite picture is found for anticyclones: for those associated with an AWB the breaking typically occurs in a similar relative position (close to their center and slightly to the

south), hence the AWB frequency PDF is high and localized, while the CWB positions are more spread around the anticyclones (largely to their east or south-east), hence the frequency PDF of CWB is lower. These results are consistent with our earlier results of Fig.3 from the manuscript.

The figure shows that composites of cyclones and anticyclones, which are customarily performed in previous studies (at least for cyclones), in fact mix between cyclonically and anticyclonically breaking storms. This motivates our further decomposition of cyclones and anticyclones into those breaking cyclonically and anticyclonically, to examine their distinct time evolution and characteristics.

We have added the above figure and discussion into the manuscript.

Intensity of systems in the composite analysis

The authors note in passing that the feature on which a composite is centered will be more intense than the other features in the composite, a well-known artefact of composites. Still, the authors use the intensity of composite systems in their subsequent arguments and conclusions. This is my main methical issue with this study. A more careful analysis and discussion of the intensity of systems is needed. This could be based on benchmark composites of cyclones and anticyclones, and again I'd suggest in this case moving material from the supplement to the main text.

We have slightly modified the text to more clearly convey that the compositing procedure will inevitably highlight the intensity of the composited feature, and that this should be considered wherever claims are made which are based on the intensity of the storm in the composite.

However, note that while we agree that the composites create this inevitable artifact, we still find it insightful to examine composites. For example, compositing over cyclones is customarily done to examine their lifecycle and time evolution. As we show here, these composites in fact mix over different types of cyclones. Here we subset these composites into cyclones participating in CWB or AWB events (where for the latter we further decomposed according to the position relative to the breaking center). Hence, it is meaningful to compare between different subsets of composites (and indeed we find interesting differences in terms of their intensities, upper level features, and subsequent evolution, specifically the relative direction of the cyclone-related flow and the upper-level RWB-related flow).

In addition, we have added to Fig.5a-e in the manuscript an additional line (thick black line) showing the corresponding PDFs of all the cyclones during AWB (previously, it was separated into those to the north and those to the south). Thus, the intensity of cyclones (Fig.5a) and anticyclones (Fig.5f) during CWB (thin black line) and AWB (thick black line) can be more directly compared.

L59:  Why do you not consider all four types of RWB? Please clarify.

For every RWB event, the RWB detection algorithm can analyse either the poleward breaking tongue, or the equatorward breaking tongue. However, these essentially yield the same RWB event. The differences arise when dividing into equatorward breaking cases occurring in the cyclonic shear (LC2) or anticyclonic shear (LC1), or poleward breaking cases occurring in the cyclonic (P1) and anticyclonic (P2) shear. In the current paper we chose not to add to the

distinction between same-sense and opposite-sense cases another distinction based on whether the RWB is occurring on the equatorward or poleward sides of the zonal flow. Hence, in practice, we take into account all for types of RWBs- LC1,LC2,P1, and P2, and only distinguish between AWB and CWB events. A further investigation into the relation between the different types of categorizations, and the synoptic scale dynamics occurring for each of the four RWB types is left for further study. This has been added to the text for clarification.

L240: I am not sure I understand. What do you mean with "just signatures of the large-scale flow"? A barotropic Rossby-wave teleconnection pattern? Can you be more explicit about your possible alternative explanation/ hypothesis, such that the result that the anomalies are averages of synoptic-scale systems becomes more significant?

What we meant is whether the SLP anomalies are mostly signatures of large-scale, slowly varying flow anomalies, or rather synoptic scale weather systems. For example, it has been suggested that the positive and negative polarities of the NAO (which are low frequency modes) are directly linked to RWB events, or more generally to changes in their frequency and location. Hence, it a priori it is not clear whether the SLP composites during RWBs are mostly signatures of high-frequency eddies (e.g., synoptic systems) or low-frequency eddies (e.g., circulation regimes). Here we show a clear relation between RWB events and migrating low-level cyclones and anticyclones, which are synoptic eddies. How these feed back into the slowly varying atmospheric modes is left for further study. This has been added to the text.

L255: In the average sense, a tripole emerges, but you do not analyze if the tripole does indeed co-occur often with AWB. Please substantiate the evidence or weaken your statement.

Following your comment, we have performed some further analysis to verify how often this tripole indeed occurs during AWB events. Overall, in ~56% of the total cases we find a tripole with a C-AC-C structure that is N-S oriented. This has been added to the manuscript.

More specifically, our results show that in ~6% of the cases, the anticyclones appear without any cyclone present, in ~23% of the cases there is only 1 cyclone present, and in ~70% of the cases, there are at least two cyclones present. Out of these 70%, in ~80% of the cases (i.e., which equals to the 56% of the total mentioned above) there is at least one cyclone to the north and one cyclone to the south of the anticyclone (co-occurring).

L259: I do not understand this paragraph. What is the additional information? Why switching back to physical space? Why do you not consider the pdf in the composite? Why now including stronger systems only? Please clarify and provide more motivation for this approach. (Furthermore, note that several of your figure references have not been resolved.)

We have added the PDFs to the relative positions as well, as the reviewer suggested, and we are now including all storms and not just the stronger ones to avoid confusion (we originally chose the stronger ones just to highlight their positions, but the results are qualitatively the same).

The geographical distribution of the North-Atlantic storms and their propagation characteristics are important for regional weather (for example in the European-Asian continent and Mediterranean region), and can influence the local distribution of temperature, precipitation and extremes. Since AWB and CWB occur in different locations over the Euro-Atlantic region (Fig.1c,d), and since the relative distribution of the storms is different in each one of the cases

(Fig.3) it is also of interest to examine where do cyclones and anticyclone reside, in physical space, during AWB and CWB events. We have also added further analysis to show the PDFs of the latitudinal and longitudinal positions of the storms (new Fig.5b,c,g,h), to discuss our findings. This has been added to the text to clarify our motivation and analysis.

L276: From our understanding of cyclones that are associated with PV streamers (AWB) one would not expect them to propagate eastwards. I therefore do not see the concept of "hindering" appropriate for these cases.

We have changed the wording and no longer use the word "hindering".

Section 4: Does the analysis in this section use the closest (anti)cyclone as described in Sect. 2? Please clarify?

Yes, this has been added to the text.

Location of upper-level anomalies poleward and equatorward of the jet.

The authors often refer to the location of anomalies relative to the jet. For a single jet, I would expect that upper-level anomalies are generated due to (synoptic-scale) deflections of the jet. In this simple case one would expect ridges to be located equatorward and troughs poleward of the (instantaneous) jet. In L309, the authors emphasize this seeming "standard" case with "importantly". How would the relative locations be reversed? Does it require a double-jet structure? Can the authors be more explicit about the significance of the relative location of the anomalies?

We have slightly changed the wording to better convey what we meant. It was important for us to note that the anomalies in these cases are in the "right side of the shear", as this influences the subsequent evolution. The reviewer's notion that ridges and troughs form mainly equatorward and poleward of the jet (respectively) due to meridional deflections of the jet is correct, but only applies under a linear approximation. Indeed, during split jets or during nonlinear wave breaking events, such simple arguments may not hold. Examples of such cases include the "opposite-sense" cases that we find in the current paper. For example, in the composites of CWB events the ridge is poleward of the jet, while in the composites of AWB events the breaking trough is between the two jets.

L366: I would always expect negative meridional velocity (northerly winds on the NH) between a ridge and trough downstream. In what sense is this velocity favored in this flow configuration?

What we meant is that an asymmetry in the meridional velocity forms since it is mainly the negative meridional wind between the ridge and the developing downstream trough that intensifies (as opposed to the positive meridional velocity between the ridge and the upstream trough). The text has been revised accordingly.

Section 5: I have difficulties to see what we have learned from the „anomalous" life cycles. The section is mostly "show and tell". In the last paragraph of section 5.1: How would the shear enforce the trough? Evidently, we'd need some equatorward displacement of the PV contour to form the trough. How is the cyclonic shear doing this? How do we see the momentum fluxes? What, quite generally, do we learn in section 5.2? The statement in the last sentence of this

section has long been realized and Mediterranean cyclones are a very active field of research. In fact, the number of „anomalous" cases is of the same order of magnitude than other events. RWB looks quite different for the. „anomalous" cases. Can you clarify what we have for RWB? That RWB occurs with two distinct "flavors"?

We have slightly changed the order of the manuscript as well as its focus. Previous section 5 is now included in Section 4. In addition, we slightly modified previous section 5.1 and 5.2 to better convey our point. The last paragraph of previous section 5.1 was changed. We know compare more directly the time evolution of anticyclones during AWB and CWB events, which was the original aim of these sections.

The purpose of sections 5.1 and 5.2 was to show that the opposite-sense composites give very different cyclone and anticyclone life-cycle and time evolutions, depending on the type of upper-level RWB they are associated with (as well as their position relative to the breaking). These different lifecycles are often missed when performing composites over all storms, since cyclones and CWB as well anticyclones and AWB are more co-located. Given that these time evolutions involve different wave-mean flow interactions and jet shifts, and different storm characteristics such as intensities, positions and displacements, it is of interest to correctly identify them. For example, as the reviewer suggested, it is already acknowledged that antecedent AWB event can influence subsequent tropical, subtropical, or Mediterranean cyclone development (which include the cyclones to the SE of AWB subset). Here the other subsets are investigated as well, with potential implications for the prediction of subsequent storm development.

Note that the RWB structures look different mostly because the centering of the composites is on different storms in each case (which are often not close to the breaking center, e.g., in the case of cyclones to the NW of AWB). To discuss different "flavours" of RWB events, it would have been more insightful to composite over RWB events depending if their closest feature is a cycone/anticyclone and their position relative to the storm (e.g., to decompose them based on new Fig.6 in the manuscript). This is however beyond the scope of the current paper.

e.g., L494, rotation of upper- and lower-level anomalies: I believe that work along the line of the following reference is relevant here. - Rivière, G., Arbogast, P., Lapeyre, G. and Maynard, K., 2012. A potential vorticity perspective on the motion of a mid-latitude winter storm. Geophysical Research Letters, 39(12).

Thank you for this comment. The above-mentioned paper discusses the poleward motion of cyclones due to nonlinear PV advection. It is relevant for the current study but not necessarily in the suggested sentence. It was added when discussing the cyclonic rotation of the cyclone and the anticyclone during CWBs (lines 415-416). We have also cited an earlier related study (Gilet et al. 2009, Nonlinear baroclinic dynamics of surface cyclones crossing a zonal jet, Journal of the Atmospheric Sciences, 66, 3021-3041.

In the suggested sentence, we have added another reference that shows the wrapping-up of cyclone composites during their time of peak intensity (Dacre et al. 2012, An Extratropical Cyclone Atlas: A Tool for Illustrating Cyclone Structure and Evolution Characteristics, Bull. Am. Meteorol. Soc., (93) 1497-1502, referring to Figure 4 in that paper).

Second last sentence in the conclusions: There is a large body of literature that examines the relation of RWB - or more generally scale interactions - and weather regimes. What specific aspects would the authors like to "examine more deeply"? What are the open questions that their approach could address? In the current version, the statement is so general that it seems rather meaningless.

Understanding the relation between real-atmosphere weather systems and RWB events can help us build a better understating of how low-frequency atmospheric flows (e.g., persistent weather regimes) interact with the short-frequency flow (e.g., storms) through RWB events. This is especially important for the North Atlantic storm tracks, which, as opposed to the idealized baroclinic wave lifecycle experiments, involves a zonally asymmetric jet.

As shown in this study, a different time evolution is found for storms in different types of breaking events, with indications of distinct interactions with the mean-flow. Moreover, storm characteristics (such as positions, propagation directions, and displacements) are found to alter significantly with the breaking type. Given that different North Atlantic weather regimes are largely characterized by different types of wave breaking events, occurring in distinct geographical positions, it is of interest to examine the three-way interaction between the storm tracks, RWBs, and the low-frequency flow representing the weather regime. This is left for further study, but initial results show that distinct and clearly preferred storm paths, associated RWB positions, and resulting interactions with the low-frequency flow can be found for different weather regimes.

An improved understanding of the relation between storms, RWB events, and weather regimes can also help us improve our understanding of and confidence in projected future circulation changes (e.g., by relating changes in the frequency and positions of RWB events, storm tracks, and the North-Atlantic jet).

A related discussion has been added to the conclusions.

Editorial comments:

Abstract

 I did not find the abstract to be informative, because I could not identify the main motivation (open question) and main results of the study (see also main comment above).

The abstract has been completely revised and now reads:

Rossby wave breaking events describe the last stage in the life-cycle of baroclinic atmospheric disturbances. These breaking events can strongly influence the large-scale circulation and are also related to weather extremes such as heat waves, blockings, and extreme precipitation events. Nonetheless, a complete understanding of the synoptic-scale dynamics involved with the breaking events is still absent. Here we examine how well do idealized life cycle experiments, which use a specified initial perturbation with a single zonal wavenumber and a prescribed simplified initial zonal jet, capture the life cycle of real-atmosphere weather systems. This is done by combining a storm-tracking technique together with a wave breaking detection algorithm, focusing on the North Atlantic. These datasets also allow us to examine whether upper-level wave breaking and low-level weather systems always occur simultaneously, and if

we can we identify preferred relations between the sign of the storm and the sign of the upper-level breaking. We find that in the North Atlantic, most storms are associated with an AWB and/or CWB at some point during their lifetime, while only few cyclones and anticyclones do not involve any upper-level wave breaking (roughly 11% and 15%, respectively). Our results imply that composites of cyclones and anticyclones involve a mixture of different types of storm life-cycles, depending on whether they involve a CWB or AWB event, as well as their position relative to the RWB center. Moreover, storm characteristics (including actual and relative positions, intensities, and displacements) differ depending on the associated breaking type. We distinguish between "same-pairing" cases (i.e., cyclones with CWB and anticyclones with AWB) and "opposite-pairing" cases (i.e., cyclones with AWB and anticyclones with CWB). Compositing the cyclones and anticyclones based on this criterion, we find that in similar-pairings the surface system is positioned so that its associated upper-level winds would enhance the breaking (the anomalous circulation is in the same direction as the background shear), but for opposite-pairings, the upper-level winds associated with the surface system do not act to enhance the breaking which occurs in the direction of the background shear. A better understanding of the different life-cycles of real-atmosphere storms and the upper-level breaking they involve is important for exploring the relation between storm tracks and slowly varying weather regimes and how it is mediated by RWB events.

"Storm"-relative

The authors describe composites centered on cyclones and anticyclones as "storm-relative". This misnomer did create some confusion while reading. I suggest finding a more appropriate wording.

We changed "storm-relative" to "storm-centered".

Use of supplementary material

For me, for a research article – such as this manuscript is – there is too much use of supplementary material. Above, I have suggested how the manuscript can be strengthened and clarified by including some material in the main text. Personally, as a reader, I am usually confused by supplementary material. On page 11, e.g., there is a whole paragraph spent on describing material in the supplement. If this is important, why not including it in the main text? Do I need to consult the figure in the supplement? Or not? As a reader, I personally would like the authors to make this decision for me. I am aware, however, that I may be a minority with this opinion.

Following this comment and your previous comments, Fig.S4 and Fig.S5 from the previous version are no longer included in neither the SI nor the manuscript. Instead, we have replaced them with new Fig.6 in the manuscript. However, we have decided to leave previous Fig.S6 and Fig.S7 (now Fig.S4 and Fig.S5) in the supplementary information. We understand the reviewer's point, but we feel these figures are not essential enough to be included in the paper, yet we prefer to leave them for the interested reader. We however tried to minimize the references to figures in the SI.

L71ff: I found this sentence hard to read. It seems that what you say is that Orlanski's results are consistent with the above, but from a PV flux perspective. I suggest simplifying the presentation here for readability (of the otherwise excellently written intro).

Yes, this is what we meant. We have slightly changed the wording to simplify readability, and the sentence now reads:

"Similar conclusions were reached from a vorticity flux perspective by \citet{Orlanski2003}, who suggested that when anticyclonic circulations are dominant, the eddy vorticity flux $v'q'$ (where $q'$ is the relative vorticity) is positive poleward of the breaking and negative equatorward of it, which acts (through ${\partial{\overline{U}} \over {\partial{t}}} \sim \overline{v'q'}$) to accelerate the zonal flow poleward of the breaking and decelerate it equatorward of it (and vice versa for the case where cyclonic circulations are dominant, see also Fig.~9 in \citealp{Orlanski2003})".

L92ff: In general, I dislike the use of parentheses to condense the presentation. (I know, I know, it is often used …). While this may be convenient for the authors, but as a reader I find it often very cumbersome to read. The specific sentence here contains many parentheses because you define also a few acronyms. Please consider rewording for improved readability. I'd further appreciate if you'd minimize use of parentheses to condense sentences throughout the text.

We have changed the wording, avoiding the parenthesis, to simplify readability. This has been also done in other places throughout the text.

L118-120: Sounds like a contradiction. Please clarify.

The above sentence no longer appears in the paper.

L233: "breaking maximum in a relatively mature and developed stage". Stage of what? Wouldn't one expect a maximum being related to a mature and developed stage?

We meant that the maximum breaking instant is identified in a relatively more mature stage of the breaking, defined here as the zonal extent of the breaking tongue.

We have changed the wording to: "We note that the RWB detection algorithm we employ here identifies the breaking maximum in a relatively more mature and developed breaking stage (measured by the spatial zonal extent of the breaking tongue), which is why…".

L281ff: Are all of the statements in this paragraph supported by a figure? Please clarify.

Following your comment, we have added four more panels to Fig.5, showing the PDFs of the longitudinal (Fig.5c,h) and latitudinal (Fig.5b,g) positions of the storms during AWB and CWB events. For reference, in all the panels showing cyclones during AWB events (red dashed and dotted dashed blue lines denoting cyclones to the north and to the south, respectively), we have also added the PDF of all the cyclones during AWB (thin black line, panels a-e).

Fig. 5 shows that there is a clear longitudinal separation between the positions of cyclones and anticyclones. During CWB, both cyclones and anticyclones are generally more upstream compared to cyclones and anticyclones during AWB (Fig.5 b,g). Latitudinally (Fig.5 c,h), cyclones to the N of AWB (Fig.5c, red dashed line) are concentrated at higher latitudes compared to cyclones during CWB (Fig.5c, thick black line), while cyclones to the S of AWB (Fig.5c, blue dotted dashed line) are mostly to their south. The latitudinal distribution of anticyclones during

the two types of RWBs (Fig.5h) is similar, but anticyclones during CWB (thick black line) are slightly more concentrated to the north (difference is statistically significant at the 5% level).

For the displacements (Fig.5d,i,e,j), the differences between cyclones to the N and S of CWB (Fig.5d,e) are large and clear, as originally discussed in the manuscript. For anticyclones, the eastward propagation (Fig.5i) is slightly larger for anticyclones during AWB compared to anticyclones during CWB (difference is statistically significant at the 5% level), while the difference in the meridional displacements (Fig.5j) is not statistically significant.

We have slightly revised the paragraph accordingly.

L330: Is this shown in Fig. S6 or in the main text? I was confused, please clarify.

The separation into time-mean and anomalous flow was shown in previous Fig.S6 in the SI, now Fig.S4. The text has been revised for clarification.

L505: Why merely?

The word merely has been omitted.

**Reviewer #3**

This manuscript presents a detailed analysis of the relationship between RWB and low-level weather systems over the N Atlantic. It is well written with very clear figures, and will be acceptable with only minor revisions. My main concern is that it is not clear what the new results are (relative existing literature) and, more importantly, what questions are being (or could be) solved with this analysis. I think there needs to be better justification for the analysis, comparison with previous studies (to highlight new aspects), and discussion of implications of the results.

Thank you for this important comment, which was shared by all reviewers. We have modified the title, abstract, introduction, summary, and main body of the text (where relevant) to better convey our motivation for the current analysis, the open questions, the novelty and insights of our results and relation to previous studies. This included restructuring of old sections 4-5 into a single section 4 and changing the terminology to clarify our analysis and the novelty of our results. We have also added many references to existing literature whenever relevant, as well as to the introduction and discussions.

The motivation for the current study is to highlight the following open questions:

- What is the relation between RWB events and low-level weather systems? For example, do RWBs and weather systems always occur simultaneously? What are the percentages of storms involved with each type of breaking, and vice versa?
- How do storm characteristics (including geographical positions, intensity, and displacements) and the composite time evolution differ, depending on the type of upper-level RWB and their position relative to the breaking?
- How well do idealized life cycle experiments, which use a specified initial perturbation with a single zonal wavenumber and a prescribed simplified initial zonal jet, capture the life cycle of real-atmosphere cyclones and anticyclones?

While examining case studies can be very insightful, some of these questions cannot be addressed based on individual cases alone. Using an automated detection algorithm of RWBs and storms can therefore supplement these existing studies and generalize their results.

In a broader sense, a better understanding of the different lifecycles of real-atmosphere storms and the upper-level breaking they involve, can be useful, for example, for studying extremes, for improving weather predictability, for exploring the relation between storm tracks and slowly varying weather regimes and how it is mediated by RWB events, and for improving our confidence in projected future circulation changes (e.g., by relating changes in the frequency and positions of RWB events, storm tracks, and the North-Atlantic jet).

All of the above discussion has been added to the text to better convey our broader motivation, and the open questions specifically addressed.

Specific Comments

Title: I think "low-level" should be included in the title, Maybe "… relationship between Rossby Wave breaking and low-level weather systems"?

This has been added as suggested. The title now reads: "The relationship between Rossby Wave Breaking events and low-level weather systems".

Abstract: There is no discussion of implications of the results in the abstract , and after reading both I was left with a "so what?" feeling. I think this needs to end with discussion of implications.

The abstract has been completely revised, hopefully to better highlight the main motivation, conclusions and novelty of our results. It now reads:

Rossby wave breaking events describe the last stage in the life-cycle of baroclinic atmospheric disturbances. These breaking events can strongly influence the large-scale circulation and are also related to weather extremes such as heat waves, blockings, and extreme precipitation events. Nonetheless, a complete understanding of the synoptic-scale dynamics involved with the breaking events is still absent. Here we examine how well do idealized life cycle experiments, which use a specified initial perturbation with a single zonal wavenumber and a prescribed simplified initial zonal jet, capture the life cycle of real-atmosphere weather systems. This is done by combining a storm-tracking technique together with a wave breaking detection algorithm, focusing on the North Atlantic. These datasets also allow us to examine whether upper-level wave breaking and low-level weather systems always occur simultaneously, and if we can we identify preferred relations between the sign of the storm and the sign of the upper-level breaking. We find that in the North Atlantic, most storms are associated with an AWB and/or CWB at some point during their lifetime, while only few cyclones and anticyclones do not involve any upper-level wave breaking (roughly 11% and 15%, respectively). Our results imply that composites of cyclones and anticyclones involve a mixture of different types of storm life-cycles, depending on whether they involve a CWB or AWB event, as well as their position relative to the RWB center. Moreover, storm characteristics (including actual and relative positions, intensities, and displacements) differ depending on the associated breaking type. We distinguish between "same-pairing" cases (i.e., cyclones with CWB and anticyclones with AWB) and "opposite-pairing" cases (i.e., cyclones with AWB and anticyclones with CWB). Compositing the cyclones and anticyclones based on this criterion, we find that in similar-pairings the surface system is positioned so that its associated upper-level winds would enhance the breaking (the anomalous circulation is in the same direction as the background shear), but for opposite-pairings, the upper-level winds associated with the surface system do not act to enhance the breaking which occurs in the direction of the background shear. A better understanding of the different life-cycles of real-atmosphere storms and the upper-level breaking they involve is important for exploring the relation between storm tracks and slowly varying weather regimes and how it is mediated by RWB events.

Conclusions:

Figure 10 should be referred to in bullets (1) and (2), or introduced in the sentence before the bullets.

This has been added as suggested.

Bullets (5) and (6) seem of different flavor to (1)-(4), and I wonder if better as discussion paragraphs.

Bullets (5) and (6) have been modified and are now discussion paragraphs as suggested.

My comment on the abstract applies for the conclusions as well.

We have rewritten the conclusions section, to incorporate the changes we have made to all other parts of the text. The reviewer is referred to the new summary in the revised version. This included a modification of the schematic figure (now figure 11) to better emphasize different potential time-evolutions of cyclones and anticyclones, depending on the RWB they are associated with (and their position relative to the RWB center).

---

## Author Response (AR2)

**Reviewer #1**

I thank the authors for taking my comments into account in their revised version.
I still have some issues with the manuscript, mainly regarding the text itself, that are listed below.
The five first concern the most "major" comments. Minor comments and technicalities follow. They are given following the order of the manuscript.

"Major" comments:

1) In lines 4-5 and 130-131, the authors mention their use of "idealized life-cycle experiments". However, no such experiments have been performed in this study. The authors only use the ERA-Interim reanalysis for their investigation. From the text in the conclusion, it seems that the authors refer here to the idealized eddy life-cycles of Thorncroft et al. (1993) but also of Davies et al. (1991). Please clarify those lines (in the abstract and in the introduction).

**We indeed referred to the idealized life-cycle studies existing in the literature, and we can see why this was confusing. The text had been revised to fix this confusion.**

2) Lines 8-9: "the sign of the storm and the sign of the upper-level breaking"
I suspect that you mean cyclone/anticyclone with "sign of the storm" and the type of wave breaking with "sign of the upper-level breaking". Please change as it is not understandable as it is.
The "storm sign" expression is also present in line 632. Please adjust.

**This has been fixed**

3) The authors seem to call storm both cyclones and anticyclones. However, a storm is a synonym of a cyclone but is not a synonym of an anticyclone. Somehow, I did not catch this in the first review. Therefore, I suggest that the authors change all occurrences of storm/storms to cyclones/anticyclones when appropriate. Here are lines where it should be replaced (I may have missed some): 9 (x2), 13, 155, 159, 220, 259, 263, 266, 294 (x2), 306, 310, 339, 353, 355, 356, 357, 404, 406, 498, 502, 503, 573, 578, 579, 598, 633, 635, 636, 649, caption of Fig. 5 (4th line), caption of Fig. 5 (3rd, 4th, and 6th lines), caption of Fig. 7 (1st line), caption of Fig. 9 (1st line), caption of Fig. 10 (1st line), caption of Fig. 11 (3rd, 4th, 9th, and 10th (x2) lines).

**We have now changed "storm" to "system" or "weather system", or to cyclone/anticyclone where appropriate, to avoid confusion (expect when referring to the Lagrangian storm-tracking algorithm, which is customarily called that way, and to "storm tracks").**

4) Figure 1 (and others): it is still not clear to me what the unit of the RWB frequency is. It does not seem to be a percentage of the time as it would exceed 100% in panel c (starting from 0.4 + 4 intervals of 0.25 => frequency > 1.4). Please clarify again this aspect.

**The units are the number of events per (two-dimensional) bin area, which is different in each case. We have added this information to the text in each relevant figure.**

5) Figure 2: the titles of the columns are wrong: UPV → PV250 and U200 → U250.
Fixed

Minor comments:

- Line 4: how well do → how well

This sentence has been reworded so this no longer appears.

- Line 24: remove "flux"

Removed

- Merge paragraph 36-46 and paragraph 47-56 as they are on the same topic. As it is, after reading the paragraph I was wondering where the poleward breaking was.

Fixed

- Line 68: relative vorticity → relative vorticity anomaly

Fixed

- Lines 84 and 262: Fra → Franzke et al.

Fixed

- Line 117: One reference missing (?)

Fixed

- Line 169: e.g.Martius → e.g., Martius

Fixed

- Line 182: at its centroid → at its centroid longitude

Fixed

- Line 185: an overturning AWB (CWB) → an AWB (CWB) overturning

Fixed

- Line 186: the area of the breaking larger than 7.10-4 (as a fraction of earth's surface area) → the area of the breaking (calculated as a fraction of earth's surface area) larger than 7 10-4

Fixed

- Line 187: defined is → defined as

Fixed

- Line 187: is taken → should be

Fixed

- Line 188: spatial extent → longitudinal width ; Please check that is what you mean

Revised

- Line 188: chosen different → different

Fixed

- Line 188: the former are → the former is

We added "events" so this is correct now.

- Line 195: centeroid → centroid

Fixed

- Line 220: ceneterd → centered

Fixed

- Lines 223-224: than 2 (in units of 10-5 s-1) → 2 10-5 s-1

Fixed

- Line 248: vorticity below and → vorticity extending below and

Fixed

- Line 256: composites → composite

Fixed

- Line 256: Fig. 2d → Fig. 2c

Fixed

- Line 256: show → shows

Fixed

- Line 257: anomaly below and → anomaly extending below and

Fixed

- Line 276: absolute values, in units of 10-5s-1 → absolute value, in units of 10-5 s-1

Fixed

- Line 278: Probability Density Functions (PDFs) → PDFs ; Because the acronym is already defined

Fixed

- Line 280: strong → intense

Fixed

- Line 286: stronger → more intense

Fixed

- Line 293: where do cyclones and anticyclone reside, in physical space → where cyclones and anticyclones reside, in the physical space

Fixed

- Line 294: remove "in each case"

Fixed

- Line 298: are generally spread more → are generally more spread

Fixed

- Line 300: are concentrated more → are more concentrated

Fixed

- Lines 303,304: more in → more frequently in

Fixed

- Line 315: add reference to Fig. 5h

Fixed

- Line 318: I believe it should be five days (from T-2 to T+2), not four

Fixed

- Line 333: number events → number of events

Fixed

- Line 345: the AWB frequency PDFs around cyclones are low → the AWB frequency around cyclones is

Fixed

- Lines 346-347: the CWB frequency PDFs are high → the CWB frequency is high

Fixed

- Line 349: the frequency PDFs of CWB are lower → the CWB frequency is low

Fixed

- Line 352-353: I do not understand this sentence. Please rewrite and skip what is in parenthesis.

Revised

- Line 362: around the cyclones → on the cyclones

Fixed

- Line 384: and low-level cyclone → and a low-level cyclone

Fixed

- Line 396: south → north

Fixed

- Line 418: anticyclone) . → anticyclone).

Fixed

- Line 577: I suggest to remove ", which are customarily done in studies (at least for cyclones),"

Removed

- Line 624: what do the authors mean with "to the west"? It looks to me on Fig. 9 of Thorncroft et al. (1993) that the cyclone intensifies while moving equatorward and slightly eastward.

We meant to the west of the cyclone (this has been added). The paragraph has been slightly revised for clarity. Note that it is the anticyclone (and not the cyclone) that is moving equatorward and easatwrad

in Thorncroft at al. 1993.

- Line 625: Which idealized simulations do the authors refer to? Thorncroft's, Davies'? Please clarify.

Added

- Line 628: I suggest to remove "Taken together".

Removed

- Line 628: from idealized life-cycle studies → from previous idealized life-cycle studies

Fixed

- Line 652: anti cyclone → anticyclone

Fixed

-References:

- Line 656: here is the failing Franzke et al. (2004) reference.

Fixed

- Line 686: Schar → Schär

Fixed

- Line 716: doi missing for Hanley and Caballero (2012)

Fixed

- Line 777: Granas → Grønås

Fixed

- DOIs missing for Methven (2015), Pfahl et al. (2015) (pages and volume also missing), Rivière et al. (2012), Shapiro and Grønås (1999).

Fixed. Note that the the latter is a book

- In many references, https://doi.org/ is doubled: https://doi.org/https://doi.org/ Please check.

Fixed

- Caption of Fig. 6: centered around → centered on ; (c) , → (c),

Fixed

- Supplement:
Again, "centered around" should be changed to "centered on" everywhere it occurs.

Fixed

**Reviewer #2**

The authors have thoroughly addressed all my previous comments and resolved all my previous concerns. I find the revisions and clarifications to the manuscript very helpful to appreciate the relevance and insight of this study much better.
I recommend accepting the manuscript for publication after noting the editorial comments below.

- L8: There is a "we" too much.
fixed
- L8-9: Reading the abstract only, it is not clear to me that future readers will understand the meaning of "sign of storm" and "sign of breaking". I suggest revising for full clarity.
revised
- L117: The "?" indicates that a reference is missing.
fixed
- I was unclear with my comment in the first review on the confusion that may arise from the terminology "storm-relative". I meant to say that I find it confusing to denote an anticyclone as a storm. Similarly, not all cyclones may constitute a storm. The authors may want to rethink this terminology.
We have changed "storm" to "system" or "weather system".

---

## Author Response (AR3)

**Response to editor's comments**

Review of the manuscript entitled "The intrinsic relationship between cyclones, anticyclones, and Rossby Wave Breakings in the North-Atlantic" by Tamarin-Brodsky and Harnik

**Dear authors,**

Thank you for thoroughly addressing the remaining reviewer concerns. The paper is now accepted for publication. When once more reading the paper, I only noted to minor technical issues which you should check prior to final publication. Kind regards Christian Grams

1. Only now I noted, that it is difficult to find in the text - except for the abtract - the geographic focus of your study. Please state again in introductory paragraph of Section 2 that the geographic focus of the study is North Atlantic (indicate exact lonlat range), perhaps repeat this in Paper outline end of Section 1, in explanation of composites Section 2.3, and when starting explaining the results in Section 3.

**This has been added as suggested**

2. Fig. 6. I still find it confusing that the shading, bold and thin contours are the same in both plots of AWB (a,b) and CWB (c,d) and not the PV / PV anomalies in the subsets shown? Or are they that similar? Please check if this is what you really want to show.

We slightly modified the titles and text for clarification. We understand that it may be confusing that we plot the same thing (except for the RWB frequency) in panels (a,c) and panels (b,d), but if we plot both RWB types on one figure in each case the figures are too dense. Note that it is not surprising that CWB/AC seems more AWB because it is a composite on AC which are more collocated with AWB (and oppositely for AWB/CY which are more collocated with CWB). That was our purposeto show that the composites of CY and AC have mixed signatures, and that the overall composite structure is dominated by the type of RWB which is more collocated with its center. Note also that this is our motivation for separating into the subsets, which is shown later in Figs. 7,9, and 10. We also slightly modified the titles so it is clearer that we are plotting "all CY" and "all AC" composites together with the relative RWB distributions (and not composites conditioned on the type of breaking).